# Recombinant origin and interspecies transmission of a HERV-K(HML-2)-related primate retrovirus with a novel RNA transport element

Zachary H Williams[1], Alvaro Dafonte Imedio[1], Lea Gaucherand[2], Derek C Lee[1], Salwa Mohd Mostafa[3], James P Phelan[2], John M Coffin[2,3], Welkin E Johnson[1]*

[1]Department of Biology, Boston College, Boston, United States; [2]Molecular Microbiology Program, Tufts University Graduate School of Biomedical Sciences, Boston, United States; [3]Department of Developmental, Molecular and Chemical Biology, Tufts University School of Medicine, Boston, United States

**\*For correspondence:**
welkin.johnson@bc.edu

**Competing interest:** The authors declare that no competing interests exist.

**Abstract** HERV-K(HML-2), the youngest clade of human endogenous retroviruses (HERVs), includes many intact or nearly intact proviruses, but no replication competent HML-2 proviruses have been identified in humans. HML-2-related proviruses are present in other primates, including rhesus macaques, but the extent and timing of HML-2 activity in macaques remains unclear. We have identified 145 HML-2-like proviruses in rhesus macaques, including a clade of young, rhesus-specific insertions. Age estimates, intact open reading frames, and insertional polymorphism of these insertions are consistent with recent or ongoing infectious activity in macaques. 106 of the proviruses form a clade characterized by an ~750 bp sequence between *env* and the 3′ long terminal repeat (LTR), derived from an ancient recombination with a HERV-K(HML-8)-related virus. This clade is found in Old World monkeys (OWM), but not great apes, suggesting it originated after the ape/OWM split. We identified similar proviruses in white-cheeked gibbons; the gibbon insertions cluster within the OWM recombinant clade, suggesting interspecies transmission from OWM to gibbons. The LTRs of the youngest proviruses have deletions in U3, which disrupt the Rec Response Element (RcRE), required for nuclear export of unspliced viral RNA. We show that the HML-8-derived region functions as a Rec-independent constitutive transport element (CTE), indicating the ancestral Rec–RcRE export system was replaced by a CTE mechanism.

## Editor's evaluation

This valuable study reports on HML-2-like proviruses found in the genomes of rhesus macaques, concluding that an HML-2 provirus underwent an ancient recombination event with a HERV-K (HML-8) related virus. The authors provide solid evidence to suggest that the recombinant retrovirus acquired a distinct mechanism for the regulation of expression of spliced and unspliced transcripts. The work should be of broad interest to virologists, as it uses molecular 'fossil-like' evidence from the genomes of modern primates to document the generation of what could be considered a new retrovirus species through recombination.

**eLife digest** Just as we study fossils to understand how animals and plants have evolved, we can study ancient viruses to understand how diseases have emerged and changed over long periods. Unlike fossils, viruses do not leave visible traces in the ground but, instead, they leave viral genes known as endogenous viral elements (or EVEs) that become permanently incorporated in their host's DNA.

HML-2s are the youngest known EVEs in the human genome. They have evolved gradually by accumulating lots of small genetic changes and no longer actively infect humans. But these virus remnants have long been suspected to play a role in prostate cancer, lupus and other human diseases. Rhesus macaques and other monkeys also have HML-2s but these are less well studied than human HML-2s. Monkeys are often used as models of human biology in research studies, therefore, understanding how HML-2s have evolved in rhesus macaques may enable researchers to establish this monkey as a model for investigating the role of HML-2s in humans.

To investigate this possibility, Williams et al. searched for HML-like EVEs in rhesus macaque genomes published in previous studies. The experiments found that, unlike human HML-2s, the macaque HML-2s underwent a sudden genetic transformation millions of years ago. They acquired a new gene from another virus that completely changed how the macaque HML-2s leave a compartment within the cells of their host that contains most of the host's genome – a key step in the life cycle of viruses.

The data also suggest that HML-2s may still be actively infecting macaques today and that these EVEs jumped from monkeys into gibbons. This is the first known example of HML-2s moving between different types of primates and it indicates there may be a risk that macaque HML-2s could infect humans.

In the future, the findings of Williams et al. may help researchers develop new approaches to treat prostate cancer and other diseases linked with HML-2s in humans.

## Introduction

Retroviruses (*Retroviridae*) are a major family of animal viruses, including human viruses capable of causing severe immunodeficiency (HIV-1 and HIV-2) or cancer (HTLV-1) (*Chermann et al., 1983*; *Gallo et al., 1984*; *Guyader et al., 1987*; *Jern and Coffin, 2008*; *Poiesz et al., 1980*). Uniquely among animal viruses, they must irreversibly integrate their genome into the host genome in order to replicate (*Brown, 1997*; *Hughes et al., 1978*). If an infected cell survives, the integrated viral genome, or provirus, will be stably inherited by daughter cells; if a germline cell is infected, the provirus may be inherited by an offspring of the host organism as a heterozygous allele, which may eventually become fixed in the descendant population (*Stoye, 2012*). Such proviruses and remnants of such proviruses, known as endogenous retroviruses (ERVs), form a major fraction of most eukaryotic genomes; in humans, ~8% of the genome is composed of such sequences (*Griffiths, 2001*). These sequences serve as a useful 'fossil record' of past viral infections, and can also play important roles in genome evolution, host–viral interactions, and disease; in some animals, such as mice and cats, reactivation and recombination of ERVs with exogenous viruses and/or other ERVs can have severe pathogenic consequences (*Bamunusinghe et al., 2017*; *Bolin and Levy, 2011*; *Johnson, 2019*; *Young et al., 2012*). No replication competent human ERV (HERV) has been identified; however, the betaretrovirus-like HERV-K group contains one subgroup, human MMTV-like-2, or HML-2 (named for its similarity to mouse mammary tumor virus (MMTV)), which includes many proviruses with full-length open reading frames (ORFs) and other intact functional elements, including two proviruses with full-length ORFs for the four core retroviral proteins Gag, Pro, Pol, and Env (*Boller et al., 2008*; *Hanke et al., 2016*; *Subramanian et al., 2011*; *Wildschutte et al., 2016*). The majority of HML-2 proviruses containing full-length ORFs are within the LTR5Hs subclade, which is the youngest HERV clade and contains many human specific and insertionally polymorphic proviruses that have not yet reached fixation in human populations (*Subramanian et al., 2011*; *Wildschutte et al., 2016*). Although none of these proviruses are replication competent, two groups have shown that a synthetic consensus young HML-2 provirus is weakly replication competent, and expression of HML-2 RNA and proteins has been observed in a number of different diseases (*Dewannieux et al., 2006*; *Garcia-Montojo et al., 2018*; *Goering et al.,*

*2015*; *Hanke et al., 2016*; *Magiorkinis et al., 2013*; *Schmitt et al., 2013*; *Wang-Johanning et al., 2014*; *Lee and Bieniasz, 2007*).

The oldest known human HML-2 proviruses have orthologous insertions in both apes and Old World monkeys (OWM), indicating that the clade originated in the common ancestor of the catarrhine primates (*Bannert and Kurth, 2006*). However, much less is known about the evolutionary history and extent of HML-2 activity in non-human primates than in humans. A recent study reported that HML-2 has more recently been actively replicating in gorillas than in humans, though it is still unclear whether that activity has continued to the present day; some chimpanzee-specific insertions have also been identified, although fewer than in humans or gorillas (*Holloway et al., 2019*; *Macfarlane and Badge, 2015*). A 2007 study identified 19 HML-2 proviruses in rhesus macaques (designated as 'RhERV-K'), including three with almost identical long terminal repeats (LTRs), implying relatively recent integration (*Romano et al., 2007*). More recently, another study identified several HML-2-like proviruses in rhesus macaques, detected expression of *env* mRNA, and identified one macaque with an Env-specific immune response (*Wu et al., 2016*). One of the proviruses identified, which the authors named SERV-K1, has full-length ORFs for *gag* and *env*, and only two single nucleotide insertions disrupting the *pro* and *pol* ORFs. SERV-K1 is not present in the rhesus reference genome, indicating that it is insertionally polymorphic, and its 5′ and 3′ LTRs are identical in sequence. Both of these characteristics indicate that SERV-K1 is a relatively young insertion (less than about 500,000 years old), and suggests that rhesus macaques could possibly harbor currently or recently active HML-2-like retroviruses. Lastly, a recent study identified SERV-K1-like proviruses in a number of OWM species, including multiple macaque species, geladas, and olive baboons; this study also reported that many of these proviruses had very similar LTRs, suggestive of recent integration (*van der Kuyl, 2021*). Overall, these studies indicate the potential for recent HML-2 activity in multiple catarrhine primate species, including rhesus macaques. Thus, a comprehensive characterization of HML-2 proviruses in the rhesus macaque reference genome would provide a valuable starting point for further investigations of this potential. In order to investigate the possibility that HML-2 viruses were recently or are currently active in rhesus macaques, we searched for HML-2 proviral sequences in the two most recent rhesus genome assemblies and performed an in-depth characterization of their structure, coding capacity, and evolutionary history, with a particular focus on SERV-K1-like proviruses. We have found that SERV-K1 is part of a large clade of rhesus proviruses that show signs of recent activity. Furthermore, this clade originated as an ancient recombinant between an HML-2-like virus and another retrovirus in the HERVK11(HML-8) clade. The recombinant retrovirus has also undergone at least one interspecies transmission event, from OWM to gibbons. Lastly, we have discovered that the region derived from the recombination with HML-8 functions as a novel constitutive RNA transport element, capable of facilitating export of unspliced and partially spliced viral RNA from the nucleus.

## Results

### Structural comparison of SERV-K1 and human HML-2

SERV-K1 is closely related to human HML-2, with Gag, Pro, Pol, and Env having 79, 90, 90, and 82% amino acid identity to the proteins of the consensus human HML-2 provirus HERV-Kcon (*Lee and Bieniasz, 2007*), respectively; however, we also noticed some major structural differences (*Figure 1A*). The LTRs are significantly reduced in length, 588 bp compared to HERV-Kcon's 968 bp, with large deletions relative to human HML-2 in the U3 region (*Figure 1C*). Also, while the majority of SERV-K1 *env* is highly similar to human HML-2 *env*, ~90% identical at the amino acid level, the final 92 residues, including the transmembrane alpha helix and cytoplasmic tail, are divergent, with only 34% amino acid identity (*Figure 1B*). This divergent region of *env* is similar to the *env* genes of another ERV group, HERVK11 HML-8 (a betaretrovirus-like ERV group distantly related to HERV-K HML-2) (*Andersson et al., 1999*; *de Parseval et al., 2003*; *Mayer, 2013*). Lastly, directly downstream of *env* there is a 726-bp region that is not homologous to any sequence in human HML-2, and that has no obvious protein coding capacity. This region (in addition to the last 25 bp of the *env* ORF) appears to be derived from an HML-8 LTR, with approximately 85% nucleotide identity to MER11A elements in the human genome. MER11A is a subtype of HERVK11 HML-8 LTRs, which are classified as MER11 elements in the RepBase database of repetitive elements (*Kojima, 2018*). This relationship suggests that the ~1000 bp region comprising the last ~275 bp of *env* and 726 bp of adjacent non-coding

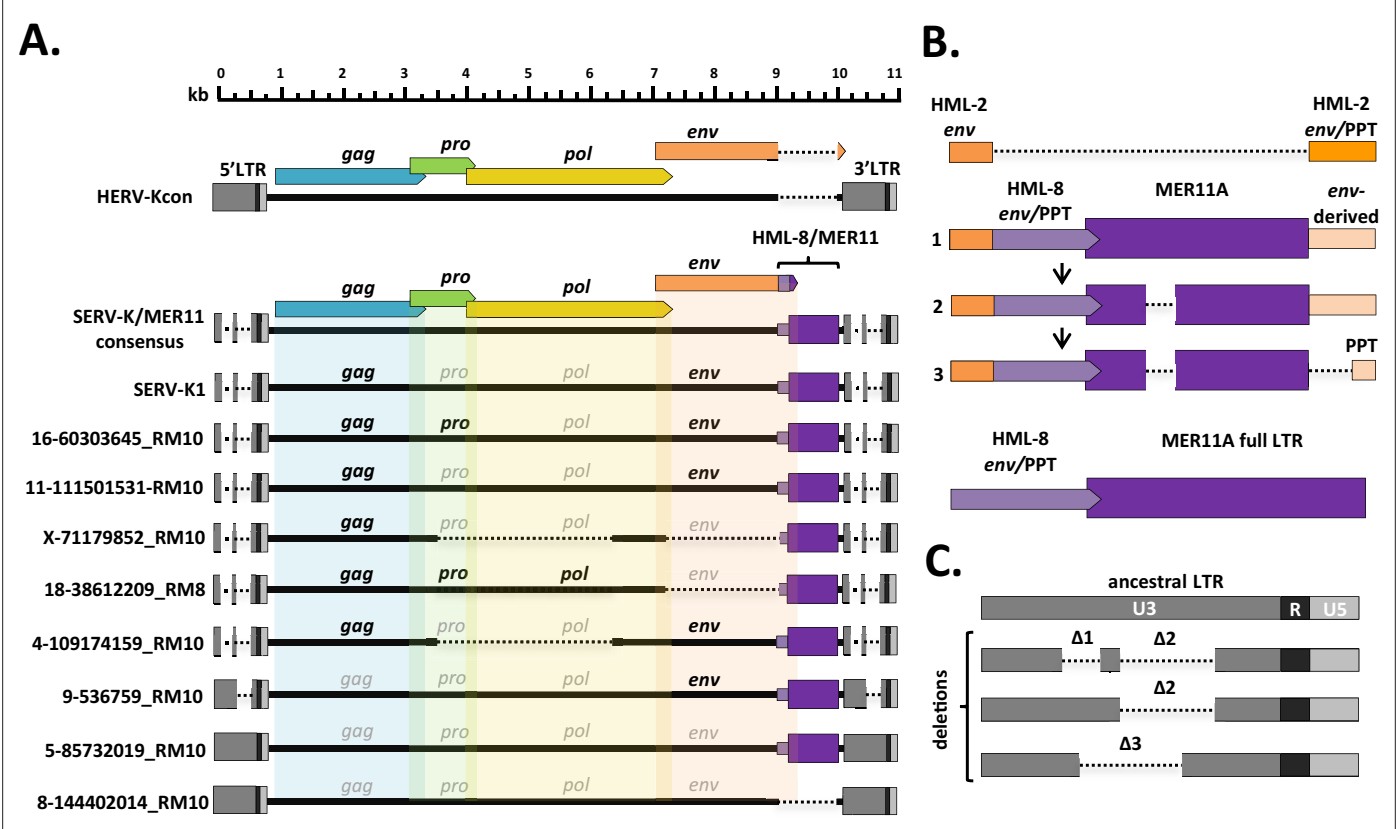

**Figure 1.** Structural features of rhesus macaque HML-2 proviruses. (**A**) Overall structure of selected rhesus proviruses compared to consensus human-specific HML-2. HERVK11 HML-8-derived region shown in purple, with light purple indicating the HERVK11-derived region of *env* and dark purple indicating the long terminal repeat (LTR)-derived MER11 element. Intact open reading frames (ORFs) shown in bold, defective or missing ORFs indicated by grayed out font. Proviruses were selected to show notable structural features such as intact ORFs, shared internal deletions, and distinct LTR types. Deletions or gaps are indicated by dotted lines. (**B**) Structural evolution of recombinant region, including parental HML-2 and HML-8 regions for comparison. 1: recombinant region in oldest proviruses, with no MER11 deletion and intact *env*-derived non-coding sequence; 2: younger recombinants with MER11 deletion; 3: youngest recombinants with MER11 deletion and deletion of *env*-derived non-coding sequence outside the polypurine tract (PPT). Bright orange = HML-2-derived *env* coding sequence; pale orange = HML-2 *env*-derived non-coding sequence (including PPT); pale purple = HML-8-derived *env* coding sequence; dark purple = HML-8 LTR-derived non-coding sequence (MER11 element). (**C**). SERV-K/MER11 LTR types, showing the accumulation of deletions in U3 in younger LTRs compared to the ancestral SERV-K/MER11 LTR. U3, R, and U5 regions of LTRs indicated by dark gray, black, and light gray coloration, respectively.

The online version of this article includes the following source data and figure supplement(s) for figure 1:

**Source data 1.** SERV-K/MER11 proviruses in rhesus macaques.

**Source data 2.** Non-recombinant HML-2-like proviruses in rhesus macaques.

**Figure supplement 1.** Origin of SERV-K/MER11 via RT-mediated recombination between co-packaged HERV-K (HML-2) and HERVK11 (HML-8) genomic RNAs.

sequence was derived by recombination between an HML-2-related retrovirus and an HML-8-like retrovirus (*Figure 1B*, *Figure 1—figure supplement 1*).

## Identification of HML-2 proviruses in rhesus macaque genomes

In order to identify and characterize other HML-2 proviruses in macaques, we searched the rheMac10 rhesus macaque reference genome assembly using the BLAST-Like Alignment Tool (BLAT) in the UCSC Genome Browser, with SERV-K1 as a query sequence (*Karolchik et al., 2011*; *Kent, 2002*; *Warren et al., 2020*). This initial search produced many hits with >95% identity to SERV-K1, all of which shared the same non-coding region in between *env* and the 3' LTR as SERV-K1, as well as many hits with lower identity, without the non-coding region. These non-coding sequences are typically annotated by RepeatMasker as MER11 elements, specifically MER11A, although some are annotated as LTR5B

elements, for reasons that are unclear to us (*Smit et al., 2015*). As a second method of identifying HML-2 proviruses, we retrieved all sequences from rheMac10 that are annotated by RepeatMasker as HERV-K internal sequence, LTRs, or MER11A, and merged adjacent annotated elements together to generate putative full proviral insertions. After filtering out solo LTRs, non-HML-2 insertions, and fragments under 2 kb, we identified 140 full-length and partial HML-2 proviruses in the rheMac10 reference genome, including 100 SERV-K1-like proviruses.

Since the rheMac8 assembly sequenced a different animal from rheMac10, we used the same methods to mine the rheMac8 genome and found 5 additional proviruses, all containing MER11 elements, for a total of 145 full-length and partial HML-2 proviruses, including 105 SERV-K1-like proviruses (*Figure 1—source data 1 and 2*; *Zimin et al., 2014*). Four additional loci were excluded as gaps were present within the assembled proviral sequences and we were not confident of proper assembly. We have assigned each proviral locus a unique identifier based on the chromosome number, leftmost coordinate of the insertion, and which genome assembly it was first identified in, for example 16-60303645_RM10 is a provirus first found in the rheMac10 assembly at chr16:60,303,645. SERV-K1 is not present in either rheMac8 or rheMac10, so we have not assigned it a new identifier, although we know that it is inserted on chromosome 12 at chr12:88,113,733. There are thus a total of 106 SERV-K1-like proviruses (including SERV-K1 itself) that are included in the following analyses.

## Structural features of macaque proviruses
### Open reading frames
We searched each provirus for the presence of full-length ORFs (defining a full-length ORF as at least 90% complete as compared to the ORFs of SERV-K1), and identified 40 proviruses with at least one full-length ORF, with a total of 31 *gag*, 17 *pro*, 6 *pol*, and 7 *env* ORFs (*Figure 1A*). No full-length ORFs were identified in non-recombinant proviruses. We did not identify any proviruses with ORFs for all four genes, however we did identify five proviruses with three intact ORFs; notably, three of these proviruses, 4-119635325_RM10, 18-38612209_RM8, and X-3424152_RM10 have intact *gag*, *pro*, and *pol*, and thus could potentially produce functional Gag-Pro-Pol polyproteins. X-3424152_RM10 *pro* is truncated by four amino acids compared to the consensus SERV-K/MER11 *pro*, but the stop codon occurs after the predicted *pro-pol* frameshift so should not affect production of full-length polyprotein.

### HERVK11 HML-8/MER11 recombination
Most of the proviruses we identified are marked by the presence of a region with homology to the HERV-K HML-8 clade of ERVs (*Figure 1B*). The HML-8 group is also known as HERVK11 in the RepBase database, with the LTRs annotated as MER11 (*Kojima, 2018*). This region begins in the TM (transmembrane) subunit of *env*, in a stretch of ~35 nucleotides with relatively high nucleotide identity between HML-2 *env* and HML-8 *env* (*Figure 1—figure supplement 1*). It then continues for approximately 1000 bp, including the transmembrane alpha helix and cytoplasmic tail of *env* TM and a 750-bp partial HML-8 LTR, annotated by RepeatMasker as a MER11A element, with the last 25 bp of *env* overlapping with the first 25 bp of the MER11 element. In the oldest provirus identified, the MER11 element is ~100 bp longer due to sequences deleted in the younger proviruses. The MER11 element is followed by a region of non-coding sequence derived from the ancestral HML-2 *env* TM. In the oldest proviruses this region is approximately 240 bp, and includes almost the entire sequence of the ancestral *env* that was replaced by HML-8 *env*; in younger proviruses most of this region has been deleted, leaving only a 42-bp sequence primarily comprising the retroviral polypurine tract (PPT), which is required for second strand synthesis during reverse transcription (*Figure 1B*). A 21-bp sequence composed entirely of purine nucleotides is immediately upstream of the MER11 element, and presumably is derived from the HML-8 PPT; it is unclear whether this sequence functions as a second PPT in the recombinant virus. The HML-8 region appears to be derived from a recombination between an HML-2 and HML-8 virus, probably between co-packaged HML-2 and HML-8 RNAs during reverse transcription.

A potential mechanism for this recombination is shown in *Figure 1—figure supplement 1*, with an initial template switch after the first strand transfer, from the HML-2 RNA in Env TM region to the HML-8 RNA in R or U3, potentially aided by microhomology. RT continues to copy the HML-8 RNA template until it reaches the HML-8 Env region, where it crosses back to HML-8 Env in a highly conserved portion of TM with ~35 bp of relatively high similarity nucleotide sequence between HML-2 and HML-8 Env. Reverse transcription continues with no further crossovers between HML-2

and HML-8. Both crossovers take place within 15–50 bp of each other on HML-2 template, resulting in an ~1100 bp insertion, including ~275 bp of Env TM and an ~850 bp partial HML-8 LTR, comprising U3 and possibly R. Thus, the initial recombinant included almost the entire nucleotide sequence of the HML-2 parental virus, with ~1100 bp of additional sequence derived from HML-8. We have tentatively named the clade of HML-2-related viruses that are descended from this initial recombinant virus SERV-K/MER11.

## Internal deletions

Although we identified many full-length or nearly full-length proviruses, a major fraction of the SERV-K/MER11 proviruses have large internal deletions (*Figure 1A*). We identified two major shared deletions; a *pro-pol* deletion (nucleotides 3384–5639 in SERV-K1) present in 32 proviruses and a *pol-env* deletion (nucleotides 6233–8152) present in 60 proviruses; 31 proviruses have both deletions (*Figure 1—source data 1*).

## LTR deletions

As previously mentioned, SERV-K1 has significantly shorter LTRs than HERV-Kcon, 588 bp compared to 968 bp. There are two major LTR deletions relative to HERV-Kcon, both in the U3 region of the LTR, ~100 and ~260 bp long, respectively, which we have designated Δ1 and Δ2 (*Figure 1C*). These deletions are found in many other SERV-K/MER11 proviruses, but not all; some proviruses have no large deletions, and have similar length LTRs to human HML-2s, others only have the Δ2 deletion, and a few proviruses have another, ~250 bp deletion distinct from the other two deletions, but overlapping them in U3 (Δ3).

## Evolutionary history of SERV-K/MER11 in OWM

To better understand the evolutionary history of HML-2 in macaques, we generated maximum likelihood trees using *pol*, *env*, and LTR sequences for all the proviruses identified. As previously mentioned, a large number of proviruses had deletions in *pol*, *env*, or both, and thus are not included in those trees. In all trees, the recombinant SERV-K/MER11 proviruses formed a well-supported monophyletic clade distinct from non-recombinant rhesus proviruses (*Figures 2 and 3*). Proviruses from the two major human HML-2 clades, LTR5Hs and LTR5A, were included for comparison in the *pol* and *env* trees, as well as MMTV as an outgroup. In both trees the LTR5Hs clade contained the nearest human HML-2 relatives to SERV-K/MER11. Many of the SERV-K/MER11 proviruses in the LTR phylogeny have very little divergence between their 5′ and 3′ LTRs; in many cases the LTRs are identical, suggestive of relatively recent integration times (*Figure 3A*). There are two distinct clades of low divergence proviruses, which we have labeled as clades 1 and 2. Notably, all of the low divergence proviruses are in the SERV-K/MER11 group; clade 1 contains the highest number of young proviruses, including SERV-K1 and other proviruses with multiple intact ORFs. We mapped the presence or absence of each LTR deletion as described in the previous section on the LTR tree; the most basal proviruses all have full length, ~1 kb LTRs, indicating that the original recombinant had intact LTRs similar in length to human HML-2s. The Δ3 and Δ2 deletions arise first, with the Δ1 deletion only showing up in the youngest, least divergent sequences; interestingly, Δ1 appears in both clades 1 and 2, despite not being present in the older members of either clade or the common ancestor of the two clades, suggesting that either Δ1 occurred twice, or entered one or both lineages through recombination.

We estimated the age of proviruses with intact LTRs using 5′–3′ LTR sequence divergence as a molecular clock (*Figure 3B*; *Johnson and Coffin, 1999*). Proviruses where the 5′ and 3′ LTRs had discordant phylogenetic histories were excluded. We compared the age estimates to those previously calculated in Holloway for human and gorilla proviruses (*Figure 3C*; *Holloway et al., 2019*); we used the same neutral substitution rate for all three species, although it is important to note that rhesus macaques and other OWM have a higher mutation rate than great apes, and thus the age estimates for rhesus proviruses are likely biased older than the true ages (*Moorjani et al., 2016*; *Palesch et al., 2018*). Other factors such as gene conversion also will greatly affect the accuracy of age estimates using this method, and thus all ages should be taken as very rough estimates (*Brown, 1997*; *Hughes and Coffin, 2005*; *Kijima and Innan, 2010*). Thirty-six of the recombinant proviruses have identical LTRs, and thus our best estimate is that they are younger than ~500,000 years old, the average time for a single nucleotide difference to occur between two 588 bp LTRs given the assumed neutral

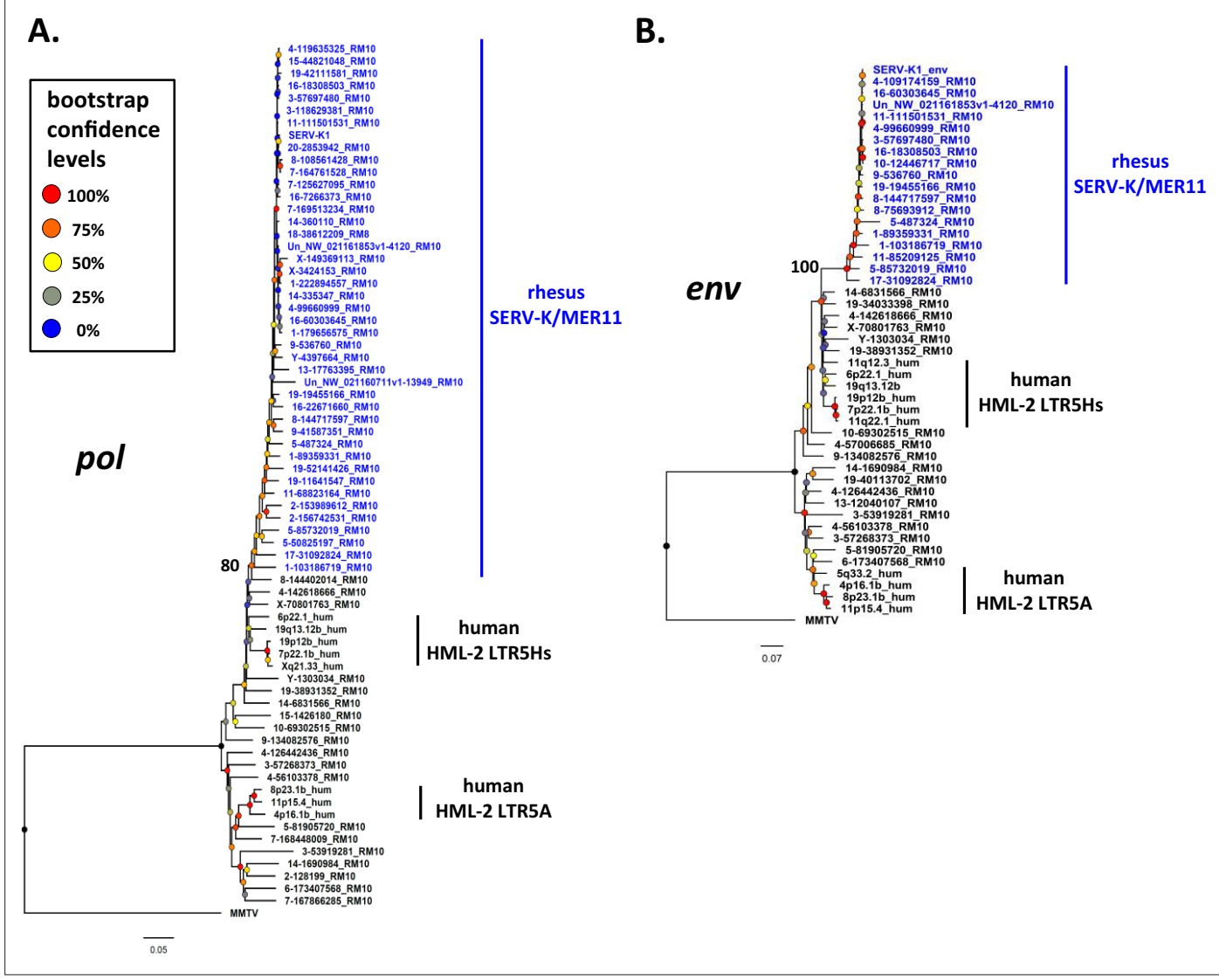

**Figure 2.** Phylogeny of SERV-K/MER11 *pol* and *env*. Maximum likelihood phylogenies for SERV-K/MER11 *pol* (**A**) and *env* (**B**) with 1000 bootstrap replicates. SERV-K/MER11 sequences are in blue. In black, sequences from non-recombinant rhesus HML-2, human HML-2 LTR5Hs, human HML-2 LTR5A included for comparison, along with MMTV *pol* or *env* as outgroups. Nodes colored according to bootstrap replicate percentages; basal node of SERV-K/MER11 clade marked with exact bootstrap value. The trees are drawn to scale, with branch lengths measured in the number of substitutions per site.

The online version of this article includes the following source data for figure 2:

**Source data 1.** *Pol* alignment.

**Source data 2.** *Env* alignment.

mutation rate; as these proviruses could have integrated at any time between 500,000 years ago and the present, we plotted the midpoint of 250,000 years. We estimate that the oldest SERV-K/MER11 provirus identified, 5-85732019_RM10, integrated ~25 million years ago. Consistent with this age, orthologous proviral insertions are present in the genomes of both cercopithecine and colobine OWM species, indicating that it integrated in a common ancestor of OWMs, prior to the cercopithecine-colobine split, between 9 and 26 million years ago (*Finstermeier et al., 2013*; *Pozzi et al., 2014*; *Springer et al., 2012*).

As younger proviral insertions are likely to be insertionally polymorphic, we screened 23 of the rhesus SERV-K/MER11 proviruses for insertional polymorphism in a panel of 14 rhesus macaque genomic DNA samples using 3 primer allele-specific polymerase chain reaction (PCR) (*Wildschutte*

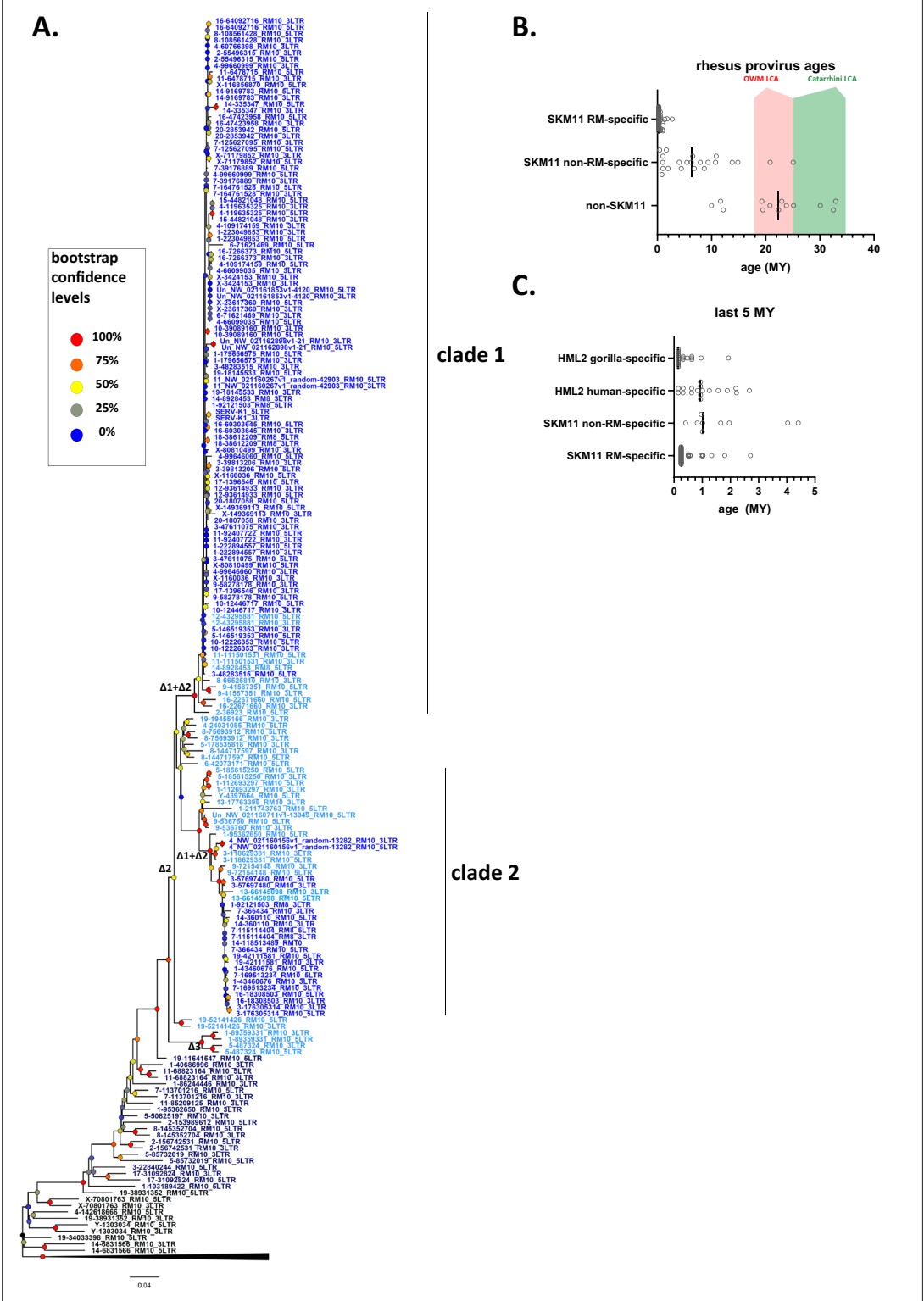

**Figure 3.** Phylogeny of SERV-K/MER11 long terminal repeats (LTRs) and LTR-based age estimates. (**A**) Maximum likelihood phylogeny of rhesus macaque HML-2 LTRs with 100 bootstrap replicates. SERV-K/MER11 clade colored blue, and shaded according to species specificity; dark blue proviruses are found in all Old World monkeys (OWM), light blue are cercopithecine specific, and royal blue are rhesus macaque specific. One clade of non-recombinant rhesus proviruses has been collapsed for clarity. Nodes colored according to bootstrap replicate values. Δ1, Δ2, Δ1 + Δ2, and Δ3 mark estimated last common ancestor for each LTR deletion or combination of LTR deletions (see *Figure 1C*). (**B**) Proviral integration times estimated using the sequence divergence between cognate 5′ and 3′ LTRs. Proviruses with no differences between their LTRs were plotted at 250,000 years. Red and

*Figure 3 continued on next page*

*Figure 3 continued*

green shaded areas mark range of estimated ages for the last common ancestors of the OWM and Catarrhine primates, respectively. SKM11 = SERV K/MER11. (**C**). Proviral integration times in the past 5 million years. Estimated ages of human and gorilla-specific HML-2s included for reference.

The online version of this article includes the following source data and figure supplement(s) for figure 3:

**Source data 1.** Long terminal repeat (LTR) alignment.

**Source data 2.** Table of estimated provirus ages.

**Source data 3.** Rhesus macaque and crab-eating macaque genomic DNA samples used for PCR screening.

**Source data 4.** Primers used for PCR screening.

**Source data 5.** Genotypes and allele frequencies of screened proviruses.

**Figure supplement 1.** Allele-specific PCR screening.

**Figure supplement 1—source data 1.** Original file for the PCR gel electrophoresis analysis in *Figure 3—figure supplement 1B*.

**Figure supplement 1—source data 2.** Original file for the PCR gel electrophoresis analysis in *Figure 3—figure supplement 1C*.

**Figure supplement 1—source data 3.** PDF containing *Figure 3—figure supplement 1B* and uncropped image of the corresponding PCR gel with relevant bands labeled.

**Figure supplement 1—source data 4.** PDF containing *Figure 3—figure supplement 1C* and uncropped image of the corresponding PCR gel with relevant bands labeled.

*et al., 2016*) with insertion specific flanking primers and a primer specific to the SERV-K/MER11 5′UTR to amplify the proviral allele and/or the pre-integration empty site; two crab-eating macaque (*Macaca fascicularis*) samples were also screened (*Figure 3—figure supplement 1*). 6/23 were determined to be insertionally polymorphic in rhesus macaques, with at least one individual heterozygous or homozygous for the empty site allele (*Figure 3—figure supplement 1*). Allele frequencies for the unfixed, insertionally polymorphic proviruses ranged from 10.7 to 70.8%, with 14-9169782 RM10 and SERV-K1 having the lowest frequencies of the screened proviruses. In contrast to SERV-K/MER11 proviruses, the non-recombinant proviruses are much older on average; the youngest non-recombinant provirus in rhesus macaques is estimated to have integrated ~10 million years ago, although orthologous insertions are present in colobine monkeys as well, suggesting that this age may be an underestimate.

Using BLAT, we determined the species distribution of each SERV-K/MER11 insertion in four other OWM reference genomes, three cercopithecine (crab-eating macaque, Anubis baboon (*Papio anubis*) and African green monkey (*Chlorocebus sabaeus*)) and one colobine (golden snub-nosed monkey (GSM; *Rhinopithecus roxellana*)), as well as the chimpanzee, gorilla, orangutan, and northern white-cheeked gibbon genomes (*Figure 3A*, *Figure 1—source data 1*, *Warren et al., 2015*; *Zhou et al., 2014*). Thirteen insertions in total were found in all OWMs, but no orthologous insertions were found in ape genomes, consistent with the recombination event that formed SERV-K/MER11 occurring after the split of apes and OWMs, 20–66 million years ago, but before the cercopithecine-colobine split (*Bergeron et al., 2021*; *Finstermeier et al., 2013*; *Pozzi et al., 2014*; *Springer et al., 2012*). Fifty-three proviruses appear to be rhesus macaque specific, as no orthologous insertion is present in the crab-eating macaque reference genome; all of these proviruses cluster within either subclade 1 or 2.

## Identification of SERV-K/MER11 proviruses in gibbons

Although we were unable to identify any orthologues of the oldest SERV-K/MER11 insertions in ape genomes, and BLAT searches using SERV-K1 as a query confirmed the absence of similar recombinant proviruses in great apes, to our surprise, a BLAT search of the northern white-cheeked gibbon (*Nomascus leucogenys*) genome (*Carbone et al., 2014*) revealed nine proviruses, with high sequence identity to SERV-K/MER11 (*Figure 4*). These proviruses have the same overall structure as SERV-K/MER11, with short LTRs, an *env* with an HML-8-derived C-terminal region followed by a MER11-derived non-coding region, and a short sequence derived from the ancestral HML-2 *env* in between the MER11 element and the 3′ LTR, containing the PPT. None of the proviruses have orthologous insertions in macaques, and they cluster within the OWM SERV-K/MER11 clade in *pol* and LTR trees, specifically clustering with a clade of SERV-K/MER11 proviruses in the GSM genome (*Figure 5A, B*). These characteristics all strongly suggest that these proviruses are the result of an interspecies transmission from an OWM to an ancestral gibbon species.

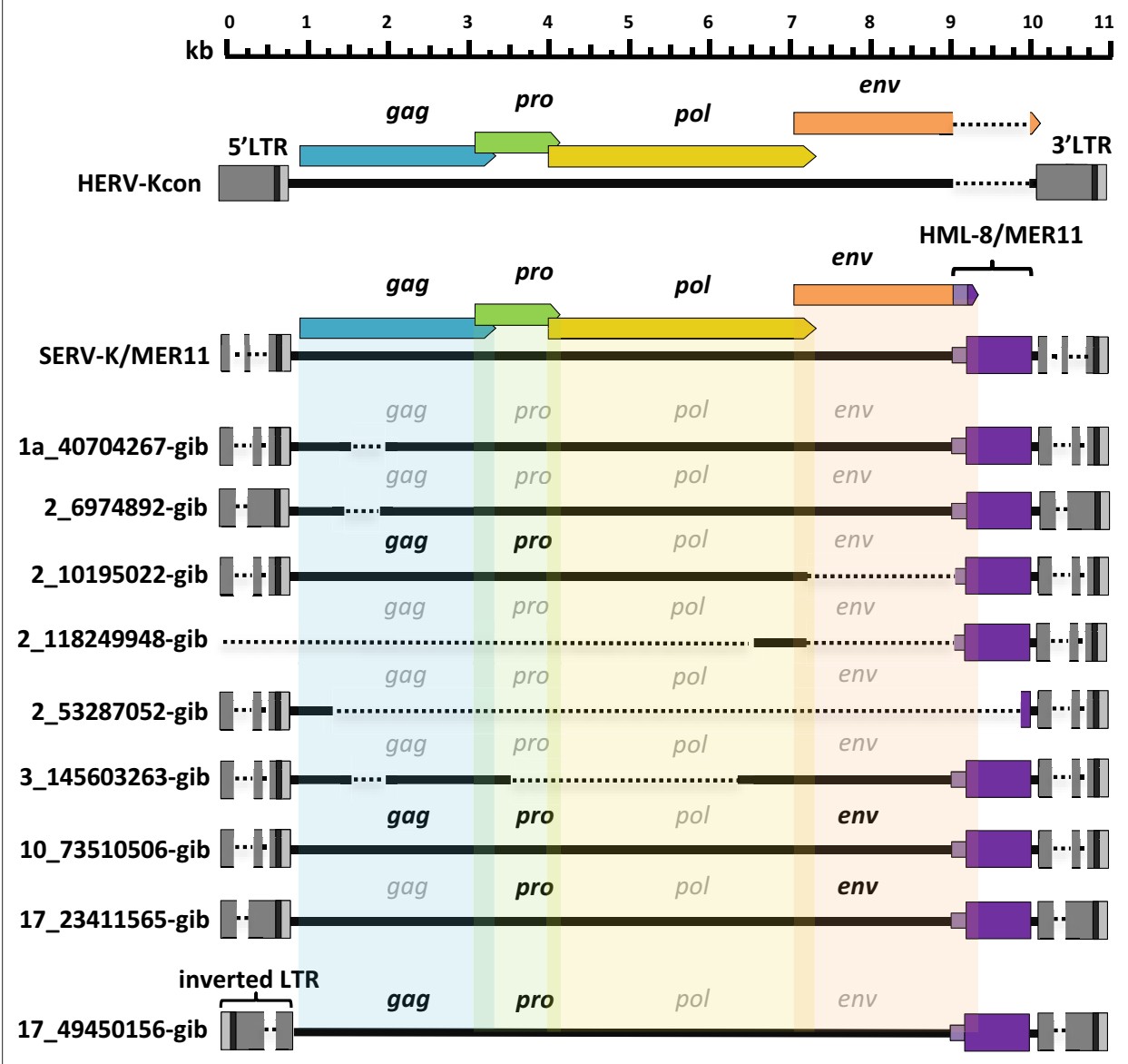

**Figure 4.** Structures of gibbon proviruses. Structure of gibbon SERV-K/MER11 provirus compared to rhesus SERV-K/MER11 and human HML-2. HERVK11 HML-8-derived region shown in purple, with light purple indicating the HERVK11-derived region of *env* and dark purple indicating the long terminal repeat (LTR)-derived MER11 element. Intact open reading frames (ORFs) shown in bold, defective or missing ORFs indicated by grayed out font. Deletions or gaps in alignment shown by dotted lines.

The online version of this article includes the following source data for figure 4:

**Source data 1.** SERV-K/MER11 proviruses in northern white-cheeked gibbon assembly.

Four of the gibbon proviruses identified have full-length ORFs for one or more genes, with a total of 3 *gag*, 4 *pro*, and 2 *env* ORFs; no intact *pol* ORFs were identified (*Figure 4*). Six of the proviruses have discordantly clustering 5′ and 3′ LTRs, suggestive of extensive inter-provirus recombination (*Figure 5B*; *Hughes and Coffin, 2005*). Interestingly, one provirus with discordantly clustering LTRs, 10_73510506-gib, appears to be the result of an inter or intrachromosomal recombination between two proviruses, resulting in a chromosomal translocation; the target site duplications (TSDs) at the ends of the provirus, which should be identical, do not match each other, and the flanking sequences map to two different human chromosomes (*Figure 5—figure supplement 1*). We identified a solo LTR at chr10:5,132,425–5,132,933 that appears to be the reciprocal recombinant locus, with flanking sequences corresponding to the other halves of the predicted chromosomal translocation,

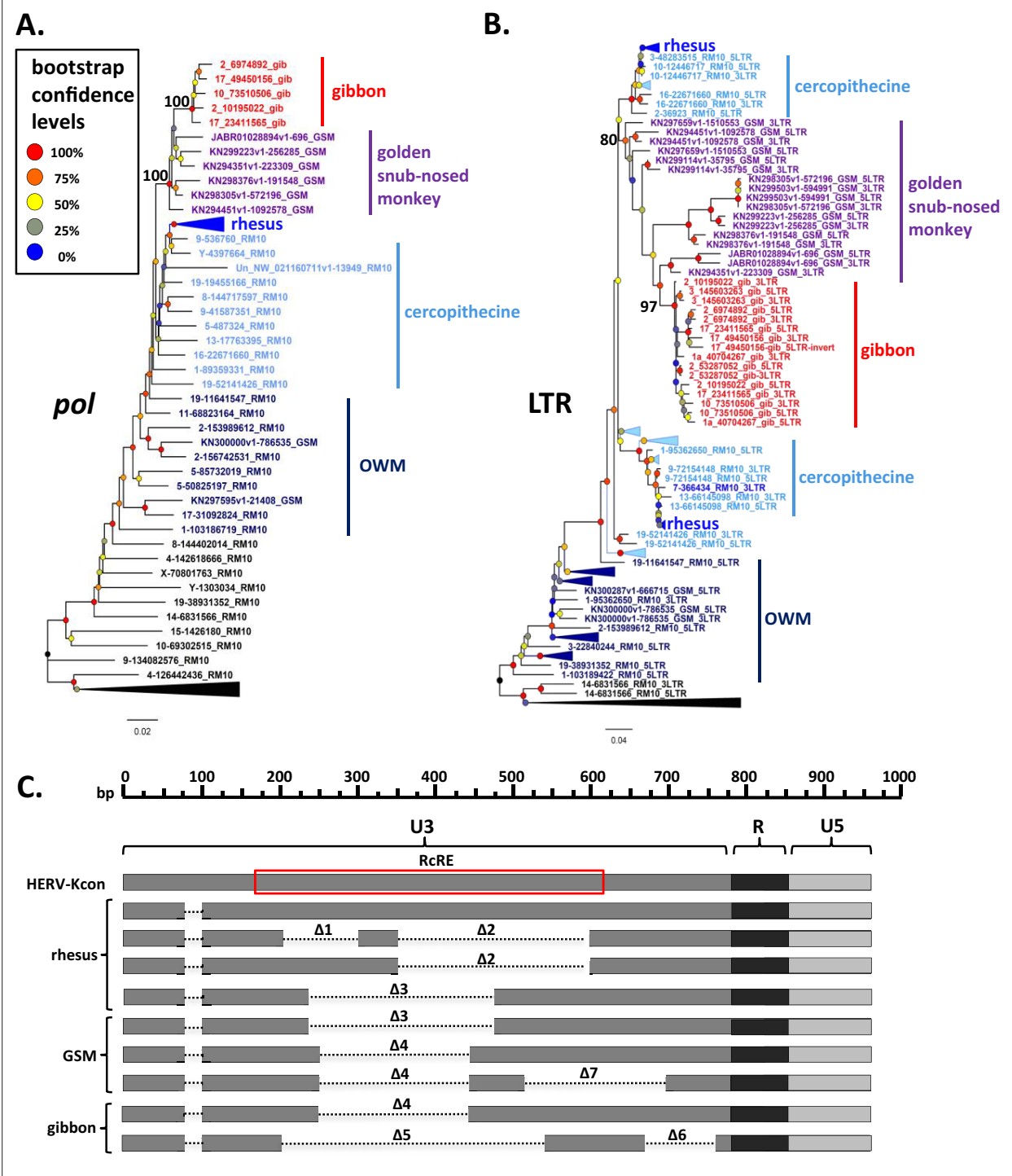

**Figure 5.** Phylogeny and long terminal repeat (LTR) deletions of gibbon and golden snub-nosed monkey (GSM) SERV-K/MER11 proviruses. (**A, B**) *Pol* and LTR maximum likelihood phylogenies of gibbon and GSM SERV-K/MER11 proviruses with 100 bootstrap replicates. Rhesus SERV-K/MER11 and non-recombinant proviruses were included for comparison; large clades of rhesus proviruses collapsed for clarity. Nodes colored by bootstrap confidence level, branches colored by provirus type and species specificity. Black = non-recombinant rhesus, dark blue = OWM SERV-K/MER11, light blue = cercopithecine specific, royal blue = rhesus specific, purple = GSM, red = gibbon. (**C**) Schematic of deletions in U3 of rhesus, GSM and gibbon SERV-K/MER11 LTRs, compared with a human HML-2 full-length 968 bp LTR (HERV-Kcon). Dotted lines mark deletions relative to the HERV-Kcon LTR. Approximate location of 433 bp HML-2 Rec Response Element (RcRE) noted by red box.

The online version of this article includes the following source data and figure supplement(s) for figure 5:

*Figure 5 continued on next page*

*Figure 5 continued*

**Source data 1.** SERV-K/MER11 proviruses in golden snub-nosed monkey assembly.

**Source data 2.** *Pol* alignment.

**Source data 3.** Long terminal repeat (LTR) alignment.

**Figure supplement 1.** 10-73510510 provirus is the result of intra- or interchromosomal recombination.

and corresponding mismatched TSDs. Two proviruses, 2_53287052-gib and 3_145603263-gib, had intact, non-discordant 5′ and 3′ LTRs and were thus amenable to age estimation, with estimated ages of ~1.2 and~2.4 million years, respectively, indicating that the interspecies transmission event occurred at least 2.4 million years ago.

## Characterization of species-specific LTR deletions in gibbon and GSM SERV-K/MER11 proviruses

As mentioned above, the LTRs of gibbon SERV-K/MER11 have deletions relative to human HML-2, similar to those of rhesus SERV-K/MER11. These deletions overlap with those found in rhesus proviruses, but are distinct from them. We identified two types of gibbon proviral LTRs: one type has a single ~200 bp deletion, Δ4, and the other type has an ~370 bp deletion followed by an ~100 bp deletion, named Δ5 and Δ6, respectively (*Figure 5C*). As previously mentioned, the closest known relatives of the gibbon SERV-K/MER11 clade are proviruses we identified in GSM; some of these proviruses have the same Δ4 deletion found in gibbons, suggesting that the viral lineage that originally crossed over into gibbons also had this deletion. In addition to GSM proviruses with a single Δ4 deletion, we identified proviruses with both Δ4 and another ~180 bp deletion, Δ7, as well as proviruses with the same Δ3 deletion found in rhesus macaques.

## Identification of a novel RNA transport element in the HML-8-derived region of SERV-K/MER11

The aforementioned LTR deletions are all found in the U3 region of the LTR. We noticed that all but one of them also overlap with the Rec Response Element (RcRE), a 433 nucleotide structured RNA element within the U3 region of human HML-2 that binds Rec, an HML-2 accessory protein that is necessary for nuclear export of unspliced and partially spliced viral mRNAs (*Figure 5C*; *Löwer et al., 1995*; *Magin-Lachmann et al., 2001*; *Magin et al., 1999*; *Yang et al., 1999*). As this RNA export process is essential for viral replication, and it seemed likely that these deletions would abrogate or abolish RcRE function, we wondered whether SERV-K/MER11 proviruses have evolved a new or altered mechanism for unspliced RNA transport. Specifically, we wondered whether the HML-8 LTR-derived MER11 region might play a role in this transport, either replacing the original RcRE, or functioning as a constitutive transport element (CTE) that does not require a viral accessory protein to function. CTEs provide the RNA transport mechanism used by Mason-Pfizer monkey virus (MPMV), avian leukosis virus, and some other simple retroviruses (*Bray et al., 1994*; *Ogert et al., 1996*; *Pessel-Vivares et al., 2015*).

We first asked whether SERV-K/MER11 proviruses still make a *rec*-like transcript. HML-2 Rec is translated from a doubly spliced RNA; the first coding exon is derived from the Env signal peptide, and the second coding exon begins near the C-terminus of Env, in an alternate reading frame (*Figure 6A*). The region containing this exon has been replaced in SERV-K/MER11 by HML-8-derived sequence; as this region is only ~50% identical to HML-2 at the nucleotide level, we were unsure whether the *rec* splice form would be preserved, or make a functional protein. As HERV-K HML-2 is known to be expressed in human embryonic stem cells and induced pluripotent stem cells (iPSCs) (*Fuchs et al., 2013*), we aligned a publically available rhesus macaque iPSC RNAseq dataset (*Fang et al., 2014*) to the SERV-K1 proviral genome using HISAT2 (*Kim et al., 2019*), and looked for reads crossing splice junctions of *rec* or other alternatively spliced RNAs using the Integrative Genomics Viewer (IGV) (*Robinson et al., 2011*). Despite the limited sequence similarity, we observed many reads corresponding to a *rec*-like splice form, with both donor and acceptor sites matching the known *rec* sites (*Figure 6*, *Figure 6—figure supplement 1*). This transcript is predicted to make a 149 aa protein, with the first 87 residues derived from the ancestral Rec protein, and the other 62 residues encoded by HML-8-derived sequence (*Figure 6*). The HML-2-derived exon of this Rec-like protein, which we have called sRec, is ~80% identical at the amino acid level to HML-2 Rec; the HML-8-derived exon,

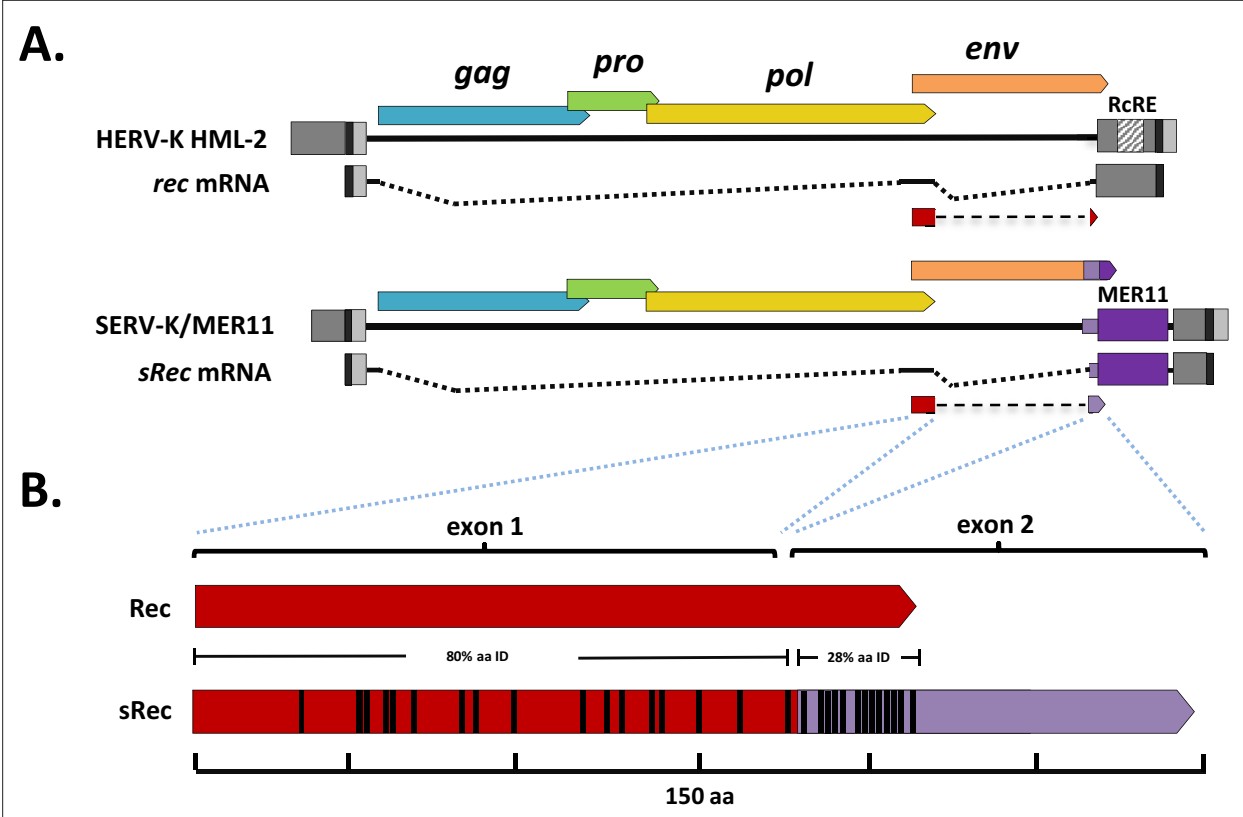

**Figure 6.** SERV-K/MER11 chimeric *rec*-like transcript and Rec response element deletions. (**A**) Comparison of human HML-2 *rec* transcript to SERV-K/MER11 *sRec* transcript. Both are doubly spliced sub-genomic transcripts. The second coding exon of *sRec* is encoded by the HML-8-derived region, in purple, with light purple denoting the HML-8 *env*-derived region, and dark purple the HML-8 long terminal repeat (LTR)-derived region. The HML-2 Rec Response Element (RcRE) in the U3 region of the LTR is also marked. (**B**) HML-2 Rec protein compared to chimeric SERV-K/MER11 sRec. Red region of sRec is homologous to Rec, with black bars denoting amino acid differences. Purple region is the 62 aa HML-8-derived exon, with black bars again showing amino acid differences.

The online version of this article includes the following figure supplement(s) for figure 6:

**Figure supplement 1.** Identification of *rec*-like transcript from rhesus macaque RNAseq data.

in contrast, is much longer than the HML-2 Rec second exon, 62 aa compared to 18 aa, and has only 28% aa identity in the first 18 residues (*Figure 6B*).

The apparent maintenance of a *rec*-like splice form suggested to us that SERV-K/MER11 may still use a Rec/RcRE-like system for unspliced RNA transport; however, the predicted RcRE region has been almost completely abrogated in the youngest SERV-K/MER11 proviruses. This led us to hypothesize that the MER11 region contains an RcRE-like element that has replaced the ancestral RcRE, that binds the sRec protein. To test this hypothesis, and the alternate hypothesis that the MER11 region contains a CTE, we adapted a dual color lentiviral reporter system designed to measure the ability of both *cis* and *trans* factors to promote nuclear export of unspliced RNAs that has been previously shown to work with the HML-2 Rec/RcRE system as well as CTEs (*Jackson et al., 2019*). This system consists of a modified NL4-3 HIV provirus that expresses eGFP from an unspliced transcript, and mCherry from a fully spliced transcript; eGFP will thus only be expressed if unspliced transcripts are exported from the nucleus (*Figure 7A*). HIV uses an accessory protein, Rev, that binds to the Rev Response Element (RRE) in unspliced and partially spliced HIV transcripts to facilitate RNA nuclear export. The reporter vector does not express Rev, and the RRE is flanked by restriction cut sites. Thus, Rev/RRE-like systems such as the HML-2 Rec/RcRE system can be tested by replacing the RRE with an RNA element of interest, and supplying the cognate viral accessory protein in trans by co-transfecting an expression vector, along with the reporter vector. CTEs can also be tested with this system, but do not require co-transfection of an accessory gene.

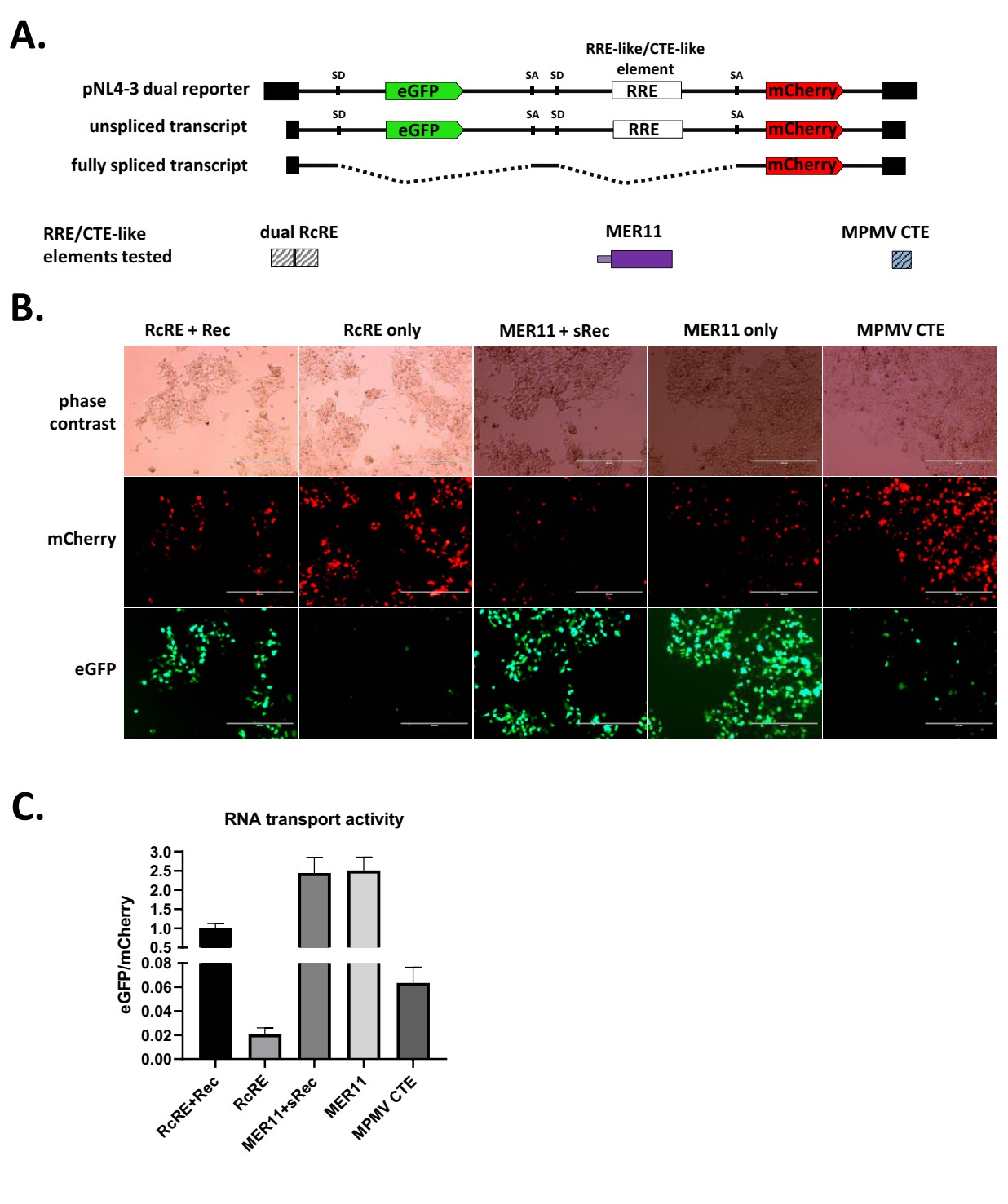

**Figure 7.** Constitutive transport element (CTE) activity of HERVK11 HML-8-derived region in SERV-K/MER11. (**A**) Schematics of dual color lentiviral reporter and transport elements tested. Base construct is an NL4-3 HIV provirus modified to express eGFP from an unspliced transcript and mCherry from a fully spliced transcript. Transport element of interest replaces RRE. SD = splice donor, SA = splice acceptor. HML-8-derived region was tested for unspliced RNA transport function by transfection with and without sRec; dual HERV-Kcon Rec Response Element (RcRE) with and without Rec was used as a control for RcRE-like activity, and MPMV CTE as a control for CTE-like activity. (**B**) Fluorescent imaging of transfected cells. mCherry = transport element-independent signal, eGFP = transport element-dependent signal. Scale bar = 400 µm. (**C**) Quantification of RNA transport activity using flow cytometry. eGFP and mCherry mean fluorescent intensity of transfected cell populations were measured, and the ratio of eGFP/mCherry plotted

*Figure 7 continued*

for each construct as a measure of RNA transport activity. The mean and standard deviation of three replicates are plotted for each condition. See *Figure 7—figure supplement 1* for gating strategy and flow dot plots for each condition.

The online version of this article includes the following source data and figure supplement(s) for figure 7:

**Source data 1.** DNA sequences of the SERV-K/MER11 consensus putative transport elements tested.

**Source data 2.** DNA sequence of consensus *sRec* open reading frame (ORF) tested for RNA transport activity.

**Source data 3.** Flow cytometry data.

**Figure supplement 1.** Constitutive transport element (CTE) activity of MER11 element with SERV-K/MER11 U3R and without HML-8 *env* region.

We tested three SERV-K/MER11 sequences, derived from a consensus clade 1 SERV-K/MER11 provirus, for RNA transport activity. The largest included the entire HML-8-derived region of *env*, the MER11 element derived from the HML-8 LTR, the trailing HML-2 *env*-derived region containing the PPT, and the predicted U3R region of the SERV-K/MER11 LTR; this sequence was synthesized and cloned into the NL4-3 lentiviral reporter to make pNL4-3(eGFP)(MER11U3R)(mCherry), using the same naming scheme as Jackson et al. From this construct, we made two smaller constructs, pNL4-3(eGFP)(MER11)(mCherry), which contains the full HML-8 *env* and MER11-derived sequence, without the PPT or U3R, and pNL4-3(eGFP)(MER11-Δ*env*)(mCherry), which contains only the LTR-derived MER11 region without the HML-8 *env*-derived sequence. We also had a consensus clade 1 SERV-K/MER11 *sRec* ORF synthesized and cloned into a pCMV expression vector to make pCMV-sRec.

These vectors were tested for RNA transport activity in HEK293T cells. We transfected each reporter construct into HEK293T cells with and without pCMV-sRec; as positive and negative controls, we used pNL4-3(eGFP)(double RcRE)(mCherry), containing two HML-2 RcRE elements in tandem, transfected with and without pCMV-Rec. As an additional positive control for CTE-like activity, we used pNL4-3(eGFP)(MPMV CTE)(mCherry), containing an MPMV CTE. We imaged the transfected cells after 48 hr, and then harvested for flow cytometry. As in Jackson et al., unspliced RNA transport activity was quantified as the ratio of eGFP mean fluorescent intensity (MFI), the transport element-dependent signal, to mCherry MFI, the transport element-independent signal. We found that all three SERV-K/MER11 constructs showed strong RNA transport activity, both with and without sRec (*Figure 7B, C*, *Figure 7—figure supplement 1*). No increase in activity was seen with the addition of sRec, and sRec did not increase transport activity when co-transfected with the RcRe reporter either (*Figure 7—figure supplement 1*). The highest eGFP/mCherry ratio was seen with the full-length MER11U3R construct, approximately 2-fold higher than the MER11 construct, which had a signal approximately 15-fold higher than MER11-Δenv. Although it shows far less RNA transport activity than the other two SERV-K/MER11 constructs, the MER11-Δenv construct has eight times higher activity than the background signal seen with RcRE in the absence of Rec, and also higher activity than the MPMV CTE positive control.

We next wanted to test the ability of the MER11-derived region to promote unspliced RNA export in the context of viral replication; to do this, we chose to use MPMV, the source of the first CTE identified, and by far the most well-studied CTE mechanism (*Bray et al., 1994*). We modified a GFP-tagged MPMV proviral construct by either deleting the MPMV wild-type CTE (wtCTE) or replacing the CTE with the MER11 element from pNL4-3(eGFP)(MER11)(mCherry). Importantly, all other functional elements were left intact in all constructs, including the MPMV PPT directly downstream of the CTE (*Figure 8A*). These modified constructs along with the parental construct, which all express eGFP in place of Env, were transfected into HEK293T cells along with a vesicular stomatitis virus glycoprotein G (VSV-G) expression vector to create VSV-G pseudotyped viral stocks. All three constructs showed strong eGFP expression in the producer cells, as expected since eGFP is expressed from the spliced *env* transcript which should express constitutively (*Figure 8—figure supplement 1*). The viral supernatants were used to infect HEK293T target cells. Infectivity was visualized by fluorescent imaging of infected cells, and quantified by measuring the percentage of eGFP expressing cells using flow cytometry (*Figure 8B, C*). As expected, MPMV with the wtCTE showed strong VSV-G-dependent infectivity, and deleting the wtCTE nearly completely abolished infectivity, despite equal or higher eGFP expression in producer cells. Replacing the wtCTE with a MER11 element restored infectivity to ~70% of wild-type levels. Thus, the SERV-K/MER11 HML-8-derived region is a bona fide CTE capable of functionally replacing the MPMV CTE during viral replication.

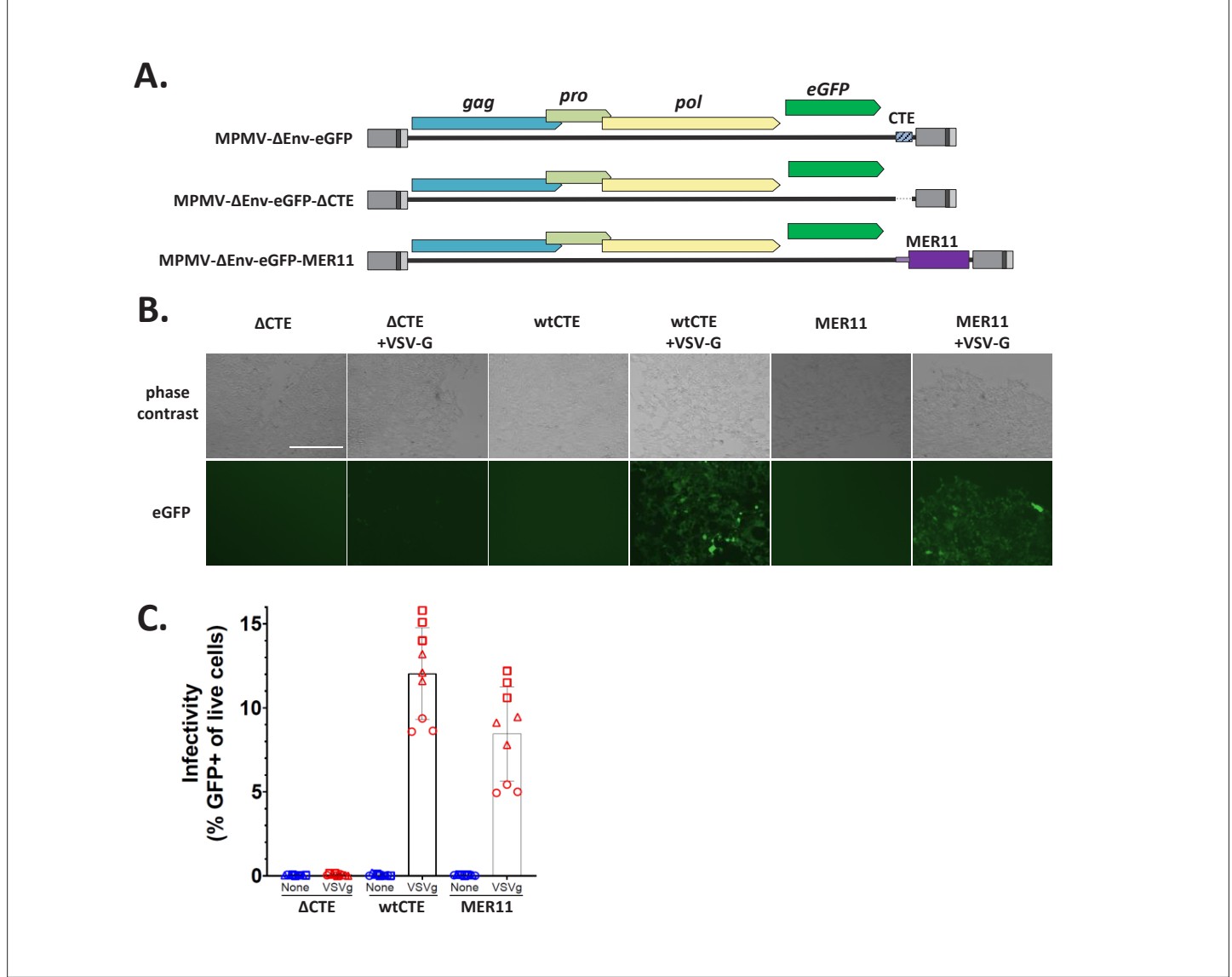

**Figure 8.** MER11 constitutive transport element (CTE) can functionally replace the MPMV CTE in the context of a single round vesicular stomatitis virus glycoprotein G (VSV-G) pseudotyped viral infection. (**A**) MPMV proviral constructs expressing eGFP in place of Env, with either the wild-type MPMV CTE (wtCTE), no CTE (ΔCTE), or the MER11 CTE replacing wtCTE (MER11), were transfected into HEK293T cells with and without VSV-G. Viral supernatants were harvested 72 hr after transfection and used to infect HEK293T target cells. Three infections were performed, each in triplicate. Infectivity was assayed after 72 hr via fluorescent imaging (**B**) with one representative image shown per condition, and flow cytometry (**C**) using % GFP expressing cells as a measure of infectivity. For images, the scale bar = 400 µm. For flow cytometry, each data point represents one experimental replicate, with different shapes corresponding to different infection rounds.

The online version of this article includes the following source data and figure supplement(s) for figure 8:

**Source data 1.** Flow cytometry data.

**Figure supplement 1.** Fluorescent imaging of producer cells for infection assay.

## Discussion

We have characterized a clade of HERV-K HML-2-related ERVs in rhesus macaques and other OWMs that include an ~750 bp non-coding region in between *env* and the 3′ LTR, derived from a recombination with a HERVK11 HML-8 retrovirus. This recombination resulted in the formation of a chimeric *env* gene, with the SU surface subunit and part of the TM transmembrane subunit derived from HML-2 *env* and the rest of TM derived from HML-8 *env*, including the transmembrane alpha helix and the cytoplasmic tail. The non-coding region is derived from the HML-8 LTR, also known as MER11.

We estimate that the recombination event which formed this clade of viruses, which we are calling SERV-K/MER11, occurred approximately 20–25 million years ago, before the split between the colobine and cercopithecine subfamilies of OWM, but after the split between apes and OWM. Consistent with this estimate, orthologs of the oldest insertions are found in species from both subfamilies of OWM, but not in any apes. This timing is supported by estimated integration times of these ancient elements based on sequence divergence of their 5′ and 3′ LTRs. The majority of the 106 recombinant proviruses so far identified in rhesus genomes are much younger, however, including two subclades of rhesus-specific insertions. Many of these young proviruses are insertionally polymorphic and present at low frequencies in macaques, and 36 of them have identical LTRs. Taken together, this evidence suggests infectious activity of HERV-K-like retroviruses in rhesus macaques within the last 500–250,000 years, and the potential for ongoing replication.

We investigated the coding capacity of these proviruses, and identified full-length ORFs for all genes, including 28 *gag*, 13 *pro*, 6 *pol*, and 7 *env* ORFs; two proviruses have intact ORFs for the full *gag-pro-pol* region and could thus potentially make functional retroviral cores, although they are lacking *env*, and so could not make infectious virus without complementation in trans from a provirus encoding a functional *env* gene. We compared the sequence of the consensus recent SERV-K/MER11 to HERV-Kcon; the overall nucleotide sequence identity of the coding region is 89.2% and amino acid identity of Gag, Pro, Pol, and Env are 78.5, 91, 91, and 82.5%, respectively. Although the sequence of these viruses has diverged, as expected for RNA viruses that have evolved separately for over 20 million years, it is likely that many of the basic features of HML-2 biology are conserved between these two viruses.

Although many of the proviruses characterized have one or more full-length ORFs, the majority of the proviruses also contain large shared deletions in the coding regions. The deletion containing proviruses are presumably not replication competent, and are reminiscent of human Type 1 HML-2s (*Löwer et al., 1993*), which contain a 292-bp *env* deletion, and other shared deletions observed in gorilla HML-2s (*Holloway et al., 2019*). Like those deletions, they are likely propagated via co-packaging with replication competent viral genomes.

In contrast to the coding regions, the LTRs of SERV-K/MER11 have undergone major structural changes, with large deletions in the U3 region. These deletions have accumulated over millions of years, with unique deletions appearing in the viral lineages in different host species, all converging on the same region of U3. Strikingly, the majority of these deletions overlap with the RcRE in human HML-2; it seems plausible that the accumulation of these RcRE deletions is causally related to the ancient recombination with HML-8 which provided the MER11 element of SERV-K/MER11. We have demonstrated that the MER11 element in a consensus young SERV-K/MER11 provirus is capable of functioning as a Rec-independent CTE, thus explaining how these proviruses are capable of functioning without an RcRE. Of the four most well-characterized betaretroviral systems (MMTV, Jaagsiekte sheep retrovirus, MPMV, and HERV-K HML-2), only one, MPMV, has a CTE system. MPMV was isolated from a rhesus macaque, and is the prototype isolate of the simian retroviruses (SRVs) of Asian macaques. Interestingly, the SRV clade is, like SERV-K/MER11, recombinant, with an *env* gene more closely related to those of gammaretroviruses than betaretroviruses (*Sonigo et al., 1986*); this raises the possibility that the SRV CTE system is also derived from a recombination event involving *env* and an adjacent non-coding sequence (the CTE).

Although these viruses appear to have lost their RcRE, they have retained a *rec*-like transcript that is predicted to produce a chimeric Rec-like protein, with a second exon derived from the recombinant HML-8-like sequence. This sRec protein does not appear to promote unspliced RNA transport, either with the human HML-2 RcRE or with SERV-K/MER11 sequences, but the conservation of this splice form suggests it may have another function in viral replication. This function may have evolved since the recombination event, or alternatively it may be that the Rec protein of human HML-2 has a second, currently unknown function. We cannot completely rule out the possibility that sRec still has some unspliced RNA transport activity, but the RcRE deletions and presence of an Rec-independent CTE in the MER11 element in combination with the lack of any promotion of unspliced RNA transport in our experiments by an sRec plasmid suggests any such activity is likely minimal. It is also as yet unknown whether the original chimeric sRec formed by the recombination event retained RNA transport activity, or whether that activity was lost over time. An immediate loss of RNA transport activity could provide a simple explanation for why these viruses switched from using a Rec–RcRE system to a CTE, if the

original recombinant HML-8-derived region already had CTE activity. However, other evolutionary scenarios remain plausible, and the hypotheses that the ancestral recombinant sRec did not have RNA transport activity and that the ancestral MER11 element had CTE activity remain unproven.

The multiple deletions in U3 centered on the RcRE region indicate that the acquisition of the MER11 CTE resulted in relaxed constraint on this region. Whether these deletions are the result of neutral evolution or were in some way adaptive is unclear, however it is worth noting that four separate lineages in three species seem to have independently acquired such deletions, which seems at least suggestive of adaptive evolution. Potential selective forces include altered U3 promoter activity (which could potentially alter viral tropism), escape from sequence-specific restriction factors such as KRAB ZNFs, or perhaps some maladaptive effect of having both a CTE and RcRE in the same virus. It is also possible that lengthening the viral genome by ~1 kb had deleterious effects on viral fitness (e.g. constraints on the maximum length of RNA that can be efficiently packaged by the capsid), and the deletions relieved this defect. In this context, it is interesting to note that SERV-K1 is only 117 bp longer than HERV-Kcon, and thus the total length of the post-recombination deletions almost matches the length of the HML-8-derived insertion.

We have tested three different MER11 constructs for CTE activity. The largest one is composed of the HML-8-derived region of *env*, the MER11 element derived from the HML-8 LTR, the PPT, and the predicted U3R region of the SERV-K/MER11 LTR; from this we made two smaller constructs, one containing the full HML-8-derived sequence, without the PPT or U3R, and one containing only the LTR-derived MER11 region. All three constructs had CTE activity, however the construct containing only the LTR-derived MER11 region had noticeably lower activity as measured by the ratio of GFP to mCherry fluorescence, which may indicate that a portion of the *env* coding region is required for full CTE function. The MER11U3R construct has somewhat higher apparent activity than the MER11-only construct, however this appears to be driven by lower mCherry expression rather than higher eGFP expression, and may be an artifact resulting from placing U3R (which contains the SERV-K/MER11 polyadenylation signal) upstream of mCherry.

It is as yet unknown whether the full MER11 element is required for CTE function, although it is notable that few deletions of the MER11 region have appeared in the SERV-K/MER11 lineage since the original recombination, which would seem to suggest that the full MER11 is necessary for viral replication, whether or not it is all part of the CTE. Interestingly, the HML-2 *env*-derived sequence in between the MER11 element and the 3′ LTR is ~240 bp long in the oldest SERV-K/MER11 proviruses, but has been reduced to a short remnant containing the PPT in the rhesus-specific clade, as one might predict since it no longer has a function as part of the *env* ORF.

The full MER11-derived region is much longer than the canonical MPMV CTE, ~1000 bp as compared to 162 bp, and longer than most other previously characterized CTE-like elements as well. However, there are other examples of CTE-like elements much longer than the MPMV CTE; the MusD transport element (MTE) is 412 bp long (*Ribet et al., 2007*), and the murine leukemia virus (MLV) post-transcriptional regulatory element (PTE) is over 1400 bp long (*Pilkington et al., 2014*), while the woodchuck hepatitis virus (WHV) post-transcriptional regulatory element or WPRE (known to enhance both export and translation of RNAs) is 592 bp long (*Donello et al., 1998*). We were not able to identify any regions in MER11 with clear sequence homology to these elements or any other published RNA export element, nor are there obvious functional motifs (e.g. the NXF1-binding loops in the MPMV CTE). However, it is predicted to have strong secondary structure (data not shown), with a complex series of stem loops extending over the entire sequence. In comparison, the MusD MTE, MLV PTE, and WHV WPRE all also have complex, multi-stem loop secondary structures (*Donello et al., 1998*; *Legiewicz et al., 2010*; *Pilkington et al., 2014*).

The above experiments are all in the context of a lentiviral reporter construct, which do not normally use a CTE, and the assay tests the ability of the element to promote viral gene expression, but does not directly test its ability to promote viral replication. Similar reporters have also previously been shown to be somewhat 'leaky', which could possibly lead to increased GFP expression via increased translation of low levels of RNA that enter the cytoplasm even without an RNA export element present (*Coyle et al., 2003*). In order to test the CTE functionality of the SERV-K/MER11 HML-8-derived region in the context of a replicating, natively CTE-using virus, we cloned this region (including both the HML-8 *env*-derived region and the MER11 element) into a GFP-tagged MPMV infectious clone in place of the original MPMV CTE, and tested this construct's ability to generate infectious virus in

a single round, VSV-G pseudotyped assay. Remarkably, this chimeric virus generated near wild-type levels of infectious virus, in contrast to an MPMV construct with the wt CTE deleted, which completely abrogated infection.

We do not yet know what RNA trafficking pathway is used by the MER11 CTE, although it could plausibly use either the NXF1/Tap pathway used by MPMV and other retroviruses (*Kang and Cullen, 1999*; *Lindtner et al., 2006*; *Sakuma et al., 2014*), or the CRM1/XPO1 pathway, used by foamy viruses (*Bodem et al., 2011*), and also by retroviruses that use Rev/RRE-like systems (*Bogerd et al., 1998*; *Yang et al., 1999*). Interestingly MLV and WHV appear to use multiple export pathways; MLV uses both NXF1- and CRM1-dependent export (*Bartels and Luban, 2014*; *Mougel et al., 2020*; *Pessel-Vivares et al., 2014*; *Pilkington et al., 2014*; *Sakuma et al., 2014*), and the WPRE has both CRM1-dependent and -independent functional elements *Popa et al., 2002*; the pathway used by MusD is still unknown (*Legiewicz et al., 2010*). It seems plausible to us that the MER11 transport element mechanism may be similarly complex.

Strikingly, the GFP/mCherry ratio of the NL4-3-derived vector containing a MER11 CTE is more than 35 times higher than that of one containing the MPMV CTE, suggesting that it may have higher RNA transport activity. However, it is important to note that the MER11 element in these reporter assays is in the context of a lentiviral genome rather than its native genomic context, which could alter its phenotype in important ways. Additionally, the readout of this assay, eGFP expression, is not a direct measure of RNA export, but rather can be affected by other factors such as RNA stability and translation efficiency. Thus at this time we cannot determine the 'true' RNA export efficiency of the MER11 element relative to the other export elements tested, and we cannot rule out the possibility that it may have other effects such as increasing translation efficiency, as has been seen for other PTEs. It is also interesting to consider whether the presence of elements that strongly promote nuclear export of intron-retaining RNAs throughout the macaque genome might have significant effects on alternative splicing of host genes, and whether HML-8 elements in the human genome might also contain CTEs that promote export of host mRNAs with retained introns. The NXF1 gene itself contains a CTE which has been shown to promote export of an alternate NXF1 isoform, and it seems plausible that ERVs might have contributed similar elements to other genes (*Li et al., 2006*; *Li et al., 2016*). The presence of a clade of very similar recombinant proviruses in the white-cheeked gibbon genome that clusters within the diversity of OWM SERV-K/MER11 sequences, as well as the absence of such proviruses from great apes strongly suggests that SERV-K/MER11 has undergone at least one inter-species transmission event. The gibbon-specific clade clusters with a group of proviruses from the GSM, a member of the Colobinae subfamily of OWM; curiously, this gibbon-GSM clade is nested within a larger clade that is otherwise specific to the Cercopithecinae subfamily. This tree structure may indicate an earlier interspecies transmission from a cercopithecine monkey to a colobine lineage ancestral to the GSM, which later crossed into gibbons; however, these deeper nodes in the phylogenies are relatively low confidence, and we cannot rule out alternative evolutionary scenarios. This is, to our knowledge, the first evidence of interspecies transmission of a virus in the HERV-K HML-2 clade, and raises a number of interesting questions, most notably, whether macaques or other OWM species still contain actively replicating SERV-K/MER11 viruses, and whether those viruses are also capable of interspecies transmission. Although the current ranges of GSM and northern white-cheeked gibbons do not overlap, the ranges of other members of their respective genera do, as do the ranges of rhesus and other macaques, consistent with the interspecies transmission or transmissions having taken place in southern Asia.

The oldest gibbon provirus we have been able to estimate an integration time for is ~2.4 million years old, indicating that the interspecies transmission to gibbons took place at least 2.4 million years ago. We were unable to estimate integration times for most of the gibbon proviruses because their 5′ and 3′ LTRs had discordant phylogenies, indicating that they have undergone some type of recombination event; one provirus appears to be derived from a crossover recombination between two separate proviruses, resulting in a reciprocal chromosomal translocation. Similar HML-2-mediated genomic rearrangements have been observed in the human genome, however no reciprocal pairs of recombinants were identified in humans (*Hughes and Coffin, 2001*). More detailed understanding of the timing and location of this transmission event may be provided by screening the genomes of other Asian primate species for SERV-K/MER11 proviruses.

It is also worth considering whether interspecies transmission has any implications for the biology of these viruses. The evidence for at least one and possibly more such transmission events is strong, and geographic proximity makes contact between the GSM and white-cheeked gibbon lineages plausible; however, both species are herbivorous, removing perhaps the most straightforward route of transmission, predation, from consideration. Additionally, the titres of reconstituted HML-2 viruses in cell culture are relatively low, which would seem to make transmission via incidental contact unlikely (*Dewannieux et al., 2006*; *Lee and Bieniasz, 2007*). It is possible that the recombinant CTE and/or LTR deletions may have altered the biology of SERV-K/MER11 compared to HERV-K HML-2 in some way that makes interspecies transmission easier.

In summary, we have characterized a clade of ERVs in rhesus macaques that may serve as a valuable model for investigating viral evolution over deep time, and as a model for HERV-K HML-2 virology and pathogenesis. This clade provides a high resolution record of the evolution of a single viral lineage going back approximately 25 million years, potentially up to the present day, that has undergone multiple significant evolutionary transitions, including recombination with divergent viral lineages and interspecies transmission to divergent host species. Rhesus macaques are a well-characterized model organism for human disease; if these viruses are still actively replicating in macaques or other OWM species, they could provide an opportunity to observe the infection dynamics of an HML-2-like virus in a natural host, which could provide insights on currently incompletely understood aspects of HML-2 biology, such as tropism, pathogenic potential, receptor usage, and mode of transmission. Lastly, we have demonstrated that SERV-K/MER11 contains a novel CTE; this provides a fascinating opportunity to investigate the evolution of a retrovirus as it shifted from using a Rev/RRE-type system to a CTE-type system, and to observe how two closely related viruses that use very different RNA trafficking mechanisms differ in their replication dynamics.

## Methods
### Genome mining
To initially identify SERV-K1-like proviruses in rhesus macaques, the BLAST-Like Alignment Tool (BLAT) on the USCG Genome Browser website (http://genome.ucsc.edu/cgi-bin/hgBlat, accessed December 3, 2019) was used to search the Mmul_8.0.1/rheMac8 and Mmul_10/rheMac10 genome assemblies, using the SERV-K1 proviral genome as a query sequence with the default search settings. As a second, complementary approach, genomic coordinates of all sequences annotated by RepeatMasker as LTR5_RM, LTR5B, HERVK-int, or MER11A (most but not all of the HML-8 LTR-derived recombinant regions are annotated by RepeatMasker as MER11A), were downloaded for the same two assemblies via the UCSC Table Browser (http://genome.ucsc.edu/cgi-bin/hgTables); coordinates of any sequences within 1 kb of each other were merged using BEDTools (*Quinlan and Hall, 2010*), and these merged coordinates were used to download candidate complete proviral sequences using UCSC Table Browser. Sequences <2 kb long were excluded, the final dataset was manually curated to remove non-HML-2 sequences that were accidentally included (primarily more distantly related ERV-K sequences such as LTR13 and HERVK11 elements that are occasionally annotated as HERVK-int by RepeatMasker), and sequences were visually inspected to remove any other errors due to misannotation.

### Allele-specific PCR
A panel of 14 rhesus macaque and 2 crab-eating macaque genomic DNA samples from the NIA Aging Cell Culture Repository at the Coriell Institute (*Figure 3—source data 3*) was screened for the presence or absence of 23 rhesus proviruses using allele-specific PCR as previously described (*Figure 3—figure supplement 1*; *Holloway et al., 2019*; *Wildschutte et al., 2016*). Primers specific to the 5′ and 3′ flanking genomic sequences of each insertion were designed using Primer3 (*Figure 3—source data 4*) and used along with a primer specific to the 5′ UTR of SERV-K/MER11 proviruses, allowing the detection of both pre-integration empty sites and proviral alleles. PCRs were performed with 100 ng of genomic DNA and Taq polymerase (Thermo Fisher).

### Multiple sequence alignment and evolutionary analysis
All alignments were generated using the MUSCLE algorithm implemented in MEGA version X (*Kumar et al., 2018*). Large insertions such as Alu and LINE-1 elements were removed from sequences to

facilitate proper alignment, and alignments were manually checked and corrected when necessary. Phylogenies were generated using the Maximum Likelihood method and Tamura-Nei model in MEGA X; 1000 bootstrap replicates were performed for *pol* and *env* trees and 100 for LTR trees. All phylogenies are drawn to scale, with branch lengths proportional to sequence divergence as measured by the number of substitutions per site. Sequence divergence data from the LTR phylogenies were used to estimate integration times as previously described (*Holloway et al., 2019*; *Subramanian et al., 2011*; *Wildschutte et al., 2016*). Briefly, the percent sequence divergence between the 5′ and 3′ LTRs of each provirus was normalized to an average divergence rate of 0.34% substitutions per site per million years, as previously measured by calculating the divergence between orthologous HML-2 proviruses in humans and chimpanzees.

## ORF identification

Full-length or nearly full-length ORFs were identified using the NCBI ORF Finder tool (https://www.ncbi.nlm.nih.gov/orffinder/), with ORFs from consensus young clade 1 SERV-K/MER11 provirus used as a reference. Genes were classified as full length or nearly full length if the first 90% of the sequence remained free of nonsense or frameshift mutations relative to the consensus gene.

## Identification of SERV-K/MER11 loci in other OWM and gibbons

To identify SERV-K-like proviruses in other primates we used BLAT searches against genome assemblies for crab-eating macaque (macFas5), Anubis baboon (papAnu4), African green monkey (chlSab2), GSM (rhiRox1), human (hg38/GRCh38), chimpanzee (panTro6), gorilla (gorGor5), orangutan (ponAbe3), and northern white-cheeked gibbon (nomLeu3), using the same approach we used for rhesus macaque, with SERV-K1 as a query. For genomes that we identified SERV-K1-like proviruses we used Table-Browser and RepeatMasker as a complementary approach to retrieve HML-2 sequences, as we did with rhesus macaques.

To determine the presence or absence of orthologous SERV-K/MER11 insertions in primates with SERV-K1-like proviruses, we used BLAT searches with proviral flanking sequences as queries. For each SERV-K/MER11 proviral insertion, we identified in rhesus macaques we retrieved the full proviral sequence from the UCSC Genome Browser, plus 1000 bp of flanking sequence upstream and downstream, and used that sequence as a BLAT query in each genome. A proviral insertion was counted as present in a species if at least one provirus-flanking sequence junction was intact in the genome assembly for that species. Proviral sequence at the same locus but without intact insertion junction sequences were not counted, as such sequences are often assembly artifacts. When 1000 bp of flanking sequence was insufficient to identify an orthologous pre-integration site, 5000 bp flanking sequences were used as queries. The most distantly related species where each proviral insertion was found is listed in the 'Oldest Common Ancestor' column of the *Figure 1—source data 1* table.

## RNAseq analysis

An Illumina HiSeq RNAseq dataset (SRA accession SRR1575130) from rhesus macaque iPSCs was downloaded from the NCBI Sequence Read Archive (*Fang et al., 2014*). Reads in FASTQ format were trimmed and quality filtered using Trimmomatic (*Bolger et al., 2014*), using an average quality score of 25 as our filter cutoff. Reads were aligned to the SERV-K1 genome with HISAT2 using the default parameters. Reads spanning the predicted HML-2 splice sites were identified by manual inspection of the alignment in the Integrative Genomics Viewer (IGV) version 2.6.2.

## Constructs and cloning for RNA trafficking assay and infection assay

The unspliced RNA trafficking assay was adapted from *Jackson et al., 2019*, and details of the dual tagged lentiviral reporter system used here are available in that paper. Briefly, a plasmid containing the full-length NL4-3 HIV genome was modified to express eGFP downstream of *gag*, from a full-length unspliced transcript, and to express mCherry from a fully spliced transcript by replacing *nef*. The construct was further modified with XmaI and XbaI restriction cut sites flanking the HIV RRE, allowing it to be replaced with other response elements or CTEs.

The Rekosh and Hammarskjold labs gave us a version of this construct, pNL4-3(eGFP)(double RcRE)(mCherry), containing two Rec Response Elements placed in tandem, replacing the HIV RRE. We had a 1545-bp putative RcRE/CTE containing sequence synthesized, consisting of the predicted 3′

end of a consensus young rhesus SERV-K/MER11 RNA genome, including the HML-8-derived portion of *env*, the MER11 non-coding region (derived from an HML-8 LTR), PPT, U3, and R, flanked by XmaI and XbaI restriction sites. Using standard cloning techniques, we replaced the double RcRE with this sequence, producing pNL4-3(eGFP)(MER11U3R)(mCherry). We then created two more constructs, first deleting the PPTU3R region to create pNL4-3(eGFP)(MER11)(mCherry), and then deleting the *env* sequence, leaving only the LTR-derived MER11 region, creating pNL4-3(eGFP)(MER11-Δenv)(mCherry). The Hammarksjold and Rekosh labs also gave us a pCMV expression vector encoding Rec; we had a consensus young sRec sequence synthesized and cloned into the same pCMV backbone as pCMV-Rec, making pCMV-sRec.

For the infection assay, we created two modified versions of pSARM4-ΔEnv-eGFP, an infectious MPMV construct with *Env* deleted and replaced by eGFP (*Song and Hunter, 2003*). We deleted the wt CTE sequence to create pSARM4-ΔEnv-eGFP-ΔCTE, and replaced the wt CTE with the full-length MER11 sequence (including the HML-8 *Env*-derived region) to create pSARM4-ΔEnv-eGFP-MER11. In both cases, the PPT in between the CTE and the 3′ LTR was preserved to allow for complete reverse transcription. These three MPMV-based viral constructs were used along with the VSV-G expressed from a pcDNA 3.1 mammalian expression vector (Invitrogen) for the infection assay (*Thomas et al., 1985*).

## Cell lines

One cell line was used in this study, HEK293T, sourced from the American Type Culture Collection (ATCC CRL-3216), authenticated via STR profiling.

## Transfection

HEK293T cells were seeded in triplicate in 12 well plates at a density of 150,000 cells/well in Dulbecco's modified Eagle's medium (Gibco) with 10% fetal bovine serum (Gibco) and incubated overnight. Cells in each well were co-transfected the next day with 0.7 μg of pNL4-3 dual reporter construct plus 0.3 μg of Rec, sRec, or empty pCMV expression vector, using the GenJet In Vitro DNA Transfection Reagent (Ver. II) protocol (SignaGen Laboratories).

## Imaging and flow cytometry

Cells were imaged 48 hr post-transfection using an EVOS digital inverted microscope (Advanced Microscopy Group). Cells were then harvested, and a FACSAria flow cytometer (BD Biosciences) was used to measure GFP and mCherry expression. Flow cytometry data analysis was performed using FlowJo version 8.7.3 (BD Biosciences, RRID:SCR_008520) and GraphPad Prism version 9.0.0 for Windows (GraphPad Software, RRID:SCR_002798). Cells were first gated to exclude dead cells and cell aggregates, and finally gated to exclude untransfected cells (GFP mCherry double negative cells). An untransfected well was first used to establish all gates which were then applied to all samples. The GFP and mCherry MFI was measured for the combined population of GFP+, mCherry+, and GFP+mCherry+ cells. The ratio of GFP MFI to mCherry MFI was calculated as a measure of unspliced RNA transport activity.

## Infection assay

HEK293T producer cells were seeded for ~90% confluence in 6-well plates 24 hr prior to transfection. pSARM4-ΔEnv-eGFP, pSARM4-ΔEnv-eGFP-ΔCTE, or pSARM4-ΔEnv-eGFP-MER11 were transfected alone or with pcDNA 3.1-VSV-G at a 3:1 ratio, according to the GenJet transfection protocol as above. Target cells were seeded for ~60% confluence in 12-well plates 24 hr prior to infection. After 72 hr, producer cells were checked for GFP fluorescence as a measure of successful transfection, and viral supernatants were harvested. For each condition, supernatants from 3 separate transfections were harvested, and for each transfection, three replicate infections were performed. Viral supernatants were centrifuged twice for 5 min at 1500 rpm to remove cells and debris. 100 μl of viral supernatant was added to each well. 72 hr after infection, cells were imaged using the EVOS digital inverted microscope as detailed above, and harvested for flow cytometry. Flow cytometry analysis was performed using a FACSAria flow cytometer as detailed above, first gating to exclude dead cells and cell aggregates, and finally gating for GFP expression, with an uninfected control used to establish the gates which were applied to all samples. Infectivity was measured as % GFP expressing cells.

## Acknowledgements

We would like to thank Marie-Louise Hammarskjold, David Rekosh, Patrick Jackson, and Sara Rasmussen at the University of Virginia School of Medicine for generously providing us with the HML-2 RcRE dual reporter construct and Rec expression construct, which we used for experiments and to create the SERV-K/MER11 reporter and expression constructs. We would also like to thank Patrick Autissier at the Boston College Flow Cytometry Core for help with flow cytometry. Work in the Johnson laboratory was supported by Boston College and by NIH grants AI083118 and AI136074; work in the Coffin laboratory was supported by NIH grant R35CA200421.

## Additional information

### Funding

| Funder | Grant reference number | Author |
|---|---|---|
| National Institute of Allergy and Infectious Diseases | AI083118 | Zachary H Williams<br>Alvaro Dafonte Imedio<br>Derek C Lee |
| National Institute of Allergy and Infectious Diseases | AI136074 | Zachary H Williams<br>Alvaro Dafonte Imedio<br>Derek C Lee |
| National Cancer Institute | R35CA200421 | Zachary H Williams<br>Lea Gaucherand<br>Salwa Mohd Mostafa<br>James P Phelan<br>John M Coffin |
| Boston College | | Welkin E Johnson |

The funders had no role in study design, data collection, and interpretation, or the decision to submit the work for publication.

### Author contributions

Zachary H Williams, Conceptualization, Data curation, Formal analysis, Supervision, Investigation, Visualization, Methodology, Writing - original draft, Writing - review and editing; Alvaro Dafonte Imedio, Formal analysis, Investigation, Visualization, Methodology, Writing - review and editing; Lea Gaucherand, Data curation, Formal analysis, Investigation, Methodology, Writing - review and editing; Derek C Lee, Resources, Investigation, Methodology; Salwa Mohd Mostafa, Resources, Investigation; James P Phelan, Investigation; John M Coffin, Supervision, Funding acquisition, Writing - review and editing; Welkin E Johnson, Conceptualization, Supervision, Funding acquisition, Writing - review and editing

### Author ORCIDs

Zachary H Williams ⓘ http://orcid.org/0000-0002-5091-2089
Lea Gaucherand ⓘ https://orcid.org/0000-0002-4477-1021
Welkin E Johnson ⓘ https://orcid.org/0000-0001-5991-5414

### Decision letter and Author response

Decision letter https://doi.org/10.7554/eLife.80216.sa1
Author response https://doi.org/10.7554/eLife.80216.sa2

## Additional files

### Supplementary files
• MDAR checklist

## Data availability

All data generated or analyzed during this study are included in the manuscript and supporting files, or are re-analyses of publically available data; source data files have been provided for Figures 1–5, 7, and 8.

The following previously published dataset was used:

| Author(s) | Year | Dataset title | Dataset URL | Database and Identifier |
|---|---|---|---|---|
| Fang R, Liu K, Zhao Y, Li H, Zhu D, Du Y, Xiang C, Li X, Liu H, Miao Z, Zhang X, Shi Y, Yang W, Xu J, Deng H | 2014 | Generation of Naïve Induced Pluripotent Stem Cells from Rhesus Monkey Fibroblasts | https://www.ncbi.nlm. nih.gov/sra/?term= SRR1575130 | NCBI Sequence Read Archive, SRR1575130 |

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
