## [Editor Report]

This valuable study reports on HML-2-like proviruses found in the genomes of rhesus macaques, concluding that an HML-2 provirus underwent an ancient recombination event with a HERV-K (HML-8) related virus. The authors provide solid evidence to suggest that the recombinant retrovirus acquired a distinct mechanism for the regulation of expression of spliced and unspliced transcripts. The work should be of broad interest to virologists, as it uses molecular 'fossil-like' evidence from the genomes of modern primates to document the generation of what could be considered a new retrovirus species through recombination.

---

## [Decision Letter]

**Decision letter after peer review:**

Thank you for submitting your article "Recombinant Origin and Interspecies Transmission of a HERV-K(HML-2)-related Primate Retrovirus With a Novel RNA Transport Element" for consideration by *eLife*. Your article has been reviewed by 2 peer reviewers, and the evaluation has been overseen by a Reviewing Editor and Sara Sawyer as the Senior Editor. The following individuals involved in the review of your submission have agreed to reveal their identity: Marie-Louise Hammarskjöld (Reviewer #1); Paul D Bieniasz (Reviewer #2).

Essential revisions (for the authors):

1) The potential of CTE function is one of the most interesting aspects of this work, but the data provided to support this argument are not conclusive. It will be essential that the authors provide additional evidence of CTE function (e.g., at least some of the RNA experiments suggested by reviewer #1). This is the sole major weakness of the manuscript but it is a substantial one and it is expected that additional experiments are performed to significantly strengthen this inference.

*Reviewer #1 (Recommendations for the authors):*

I consider the bioinformatic analysis presented in this manuscript to be very interesting and convincing. It clearly forms a basis for further studies of HERV-K-like viruses in rhesus macaques and other OWM species. Importantly, it suggests the ongoing activity of these viruses and maybe even ongoing infections. However, this remains speculative in the absence of data to show cytoplasmic RNA (particularly unspliced full-length RNA) and HERV protein expression. Such data would make the manuscript much more interesting.

To show that the SERV-K/MER11 proviruses indeed contain a functional bona fide CTE, the authors will have to directly demonstrate that completely unspliced GFP mRNA is exported from the nucleus. This could be done by analyzing total and cytoplasmic RNA and showing a direct effect on export. This should be complemented by an analysis of proviral RNA (total and cytoplasmic) from cells that are transcribing one or more of these proviruses. This will serve to verify that unspliced full-length SERV mRNA is exported from the nucleus when the proposed CTE is present. They should also further map the proposed CTE to obtain a "minimal" functional element and determine whether this element interacts with Nxf1. Furthermore, they should analyze whether the element shows any sequence similarity to previously identified CTEs.

*Reviewer #2 (Recommendations for the authors):*

One reference is erroneously cited : (Young N L and Bieniasz, 2007) should be (Lee Y N and Bieniasz 2007). – it's an Asian name that has apparently been rearranged.

---

## [Author Response]

Essential revisions (for the authors):1) The potential of CTE function is one of the most interesting aspects of this work, but the data provided to support this argument are not conclusive. It will be essential that the authors provide additional evidence of CTE function (e.g., at least some of the RNA experiments suggested by reviewer #1). This is the sole major weakness of the manuscript but it is a substantial one and it is expected that additional experiments are performed to significantly strengthen this inference.

To address this concern, we chose a direct, virological test of the ability of our putative CTE-like element to promote viral replication. To do this, we used Mason-Pfizer monkey virus (MPMV), which is the canonical CTE-using virus used by Dr. Hammarskjold and her colleagues to discover the prototypical CTE. In addition, MPMV is also a primate betaretrovirus, and thus belongs to the same genus as the SERV-K elements described in the manuscript. Specifically, we show that by adding the HML-8-derived region of SERV-K/MER11 to an MPMV provirus with its original CTE deleted, we can rescue viral infectivity to near wild type levels. As nuclear export of full length viral RNA is absolutely required for retroviral infectivity, we believe this functional result constitutes direct proof that the element we described must provide viral RNA nuclear export activity.

Reviewer #1 (Recommendations for the authors):I consider the bioinformatic analysis presented in this manuscript to be very interesting and convincing. It clearly forms a basis for further studies of HERV-K-like viruses in rhesus macaques and other OWM species. Importantly, it suggests the ongoing activity of these viruses and maybe even ongoing infections. However, this remains speculative in the absence of data to show cytoplasmic RNA (particularly unspliced full-length RNA) and HERV protein expression. Such data would make the manuscript much more interesting.To show that the SERV-K/MER11 proviruses indeed contain a functional bona fide CTE, the authors will have to directly demonstrate that completely unspliced GFP mRNA is exported from the nucleus. This could be done by analyzing total and cytoplasmic RNA and showing a direct effect on export. This should be complemented by an analysis of proviral RNA (total and cytoplasmic) from cells that are transcribing one or more of these proviruses. This will serve to verify that unspliced full-length SERV mRNA is exported from the nucleus when the proposed CTE is present. They should also further map the proposed CTE to obtain a "minimal" functional element and determine whether this element interacts with Nxf1. Furthermore, they should analyze whether the element shows any sequence similarity to previously identified CTEs.

Rather than measure RNA export biochemically, we chose to approach the question of proving CTE function virologically. We believe that the infectivity of the chimeric viruses we have generated shows conclusively that the HML8-derived region of SERV-K/MER11 must have CTE-like activity. We agree that determining the minimal functional CTE and the pathway or pathways used by this CTE are questions of great interest, and are actively working to answer these questions, however they are not within the scope of this study, and will require significant, additional work. We believe the reviewer will find that the discussion is now appropriately cautious in its claims and acknowledgement of the additional work that needs to be done to define the role or roles this element plays.

Reviewer #2 (Recommendations for the authors):One reference is erroneously cited : (Young N L and Bieniasz, 2007) should be (Lee Y N and Bieniasz 2007). – it's an Asian name that has apparently been rearranged.

We thank the reviewer for catching this error! We have corrected the citation to Lee et al.